# SAC-Diff: A Scan-Aware Consistency-Enhanced Diffusion Framework for Unsupervised Chest CT Anomaly Detection

**Xinyuan Zheng**[1,2] (iD)                XINYUAN.ZHENG@YALE.EDU

**Yoshihisa Shinagawa**[1]      YOSHIHISA.SHINAGAWA@SIEMENS-HEALTHINEERS.COM

**Sepehr Farhand**[1]         SEPEHR.FARHAND@SIEMENS-HEALTHINEERS.COM

**Chi Liu**[2]                      CHI.LIU@YALE.EDU

**Gerardo Hermosillo Valadez**[1]   GERARDO.HERMOSILLOVALADEZ@SIEMENS-HEALTHINEERS.COM

**Xueqi Guo**[1] (iD)          XUEQI.GUO@SIEMENS-HEALTHINEERS.COM

[1] *Siemens Medical Solutions USA, Inc., Malvern, PA, USA*

[2] *Department of Biomedical Engineering, Yale University, New Haven, CT, USA*

**Editors:** Accepted for publication at MIDL 2026

## Abstract

Anomaly detection in medical imaging is important but challenging due to diverse and imbalanced pathologies. Supervised methods rely on large annotated datasets and generalize poorly to unseen conditions. Unsupervised generative methods, especially diffusion models, can learn normal anatomy and detect outliers, but often hallucinate because of the Gaussian noise design and insufficient anatomical guidance. To address these challenges, we propose **SAC-Diff**, a **S**can-**A**ware **C**onsistency-Enhanced **Diff**usion framework for unsupervised anomaly detection in automated lung disease screening using chest CT. SAC-Diff adopts simplex noise for detail-preserving diffusion perturbation, integrates scan awareness via (A) subject-aware anatomical priors into conditional diffusion and (B) background-aware masking for scan-specific variations and heterogeneous lung anomalies, and enhances robustness by enforcing consistency and quantifying uncertainty through multi-sample ensembling. We evaluate SAC-Diff on two diseased datasets with various anomalies, COVID-19 and interstitial lung disease (ILD), and observe substantial improvements over prior methods. On COVID-19, SAC-Diff achieves an IoU of 0.39 (+3.75% improvement compared to existing methods) and Dice of 0.53 (+2.99%); on ILD, it improves IoU to 0.31 (+74.45%) and Dice to 0.44 (+60.40%). Our results demonstrate promise toward robust and annotation-free CT anomaly detection in hospital deployment.

**Keywords:** computer-aided diagnosis, unsupervised anomaly detection, CT

## 1. Introduction

Anomaly detection (Pang et al., 2021; Ruff et al., 2021; Samariya and Thakkar, 2023; Liu et al., 2024) is a fundamental challenge due to the inherent scarcity of well-annotated anomalous data, particularly within the domain of medical imaging (Wolleb et al., 2022; Huang et al., 2023; Tschuchnig and Gadermayr, 2022). This scarcity originates from the low prevalence of certain pathological conditions, the high cost of annotation that requires expert radiological input, and strict privacy restrictions that limit data availability. Consequently, training robust and generalizable models in this context remains a nontrivial task.

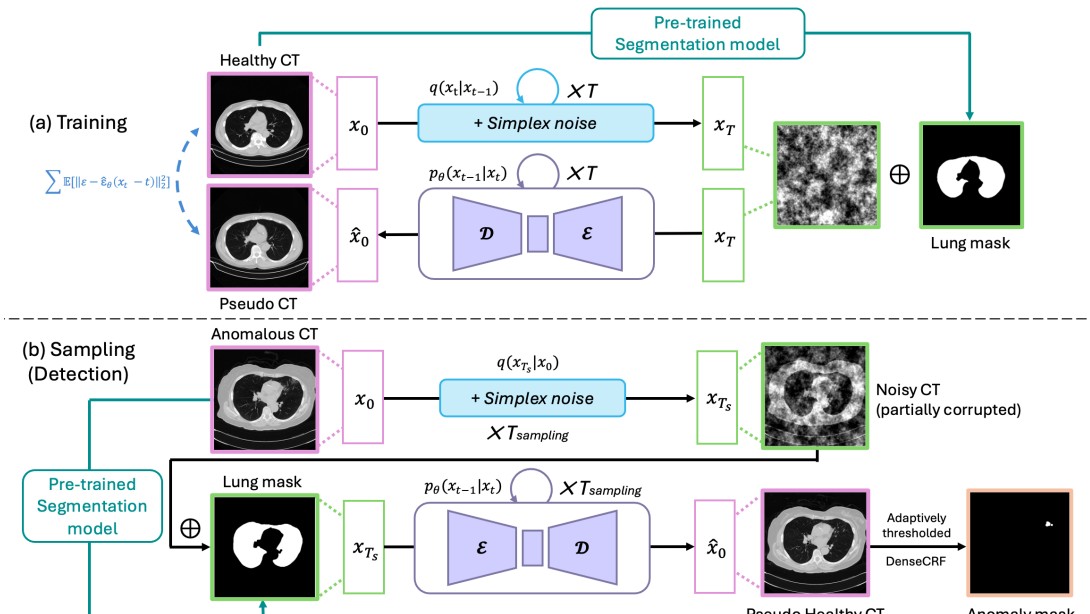

Figure 1: SAC-Diff training and detection workflow. (a) During training, healthy CT scans are diffused for $T$ steps by adding simplex noise until a completely corrupted image is obtained. A pre-segmented lung mask is incorporated as anatomical guidance, and the base model (residual UNet) is then trained to denoise these corrupted scans and reconstruct pseudo-healthy CTs. (b) At inference time, query CT scans are diffused for $T_s$ steps to obtain a partially corrupted representation. This noisy CT, together with the lung mask as anatomical guidance, is subsequently denoised by the trained network to generate a pseudo-healthy reconstruction. The anomaly mask is derived from the difference MSE map through a scan-aware background-adaptive masking strategy.

To address the scarcity of abnormal medical data, previous approaches involve artificially injecting lesion-like regions into normal images (Pezeshk et al., 2017; Salem et al., 2019; Huang et al., 2022) or synthesizing pathological data to increase the scale of the training set (Mok and Chung, 2019; Abdelhalim et al., 2021; Basaran et al., 2024; Li et al., 2020). However, these approaches often fail to capture the complex anatomical variability present in real pathological cases. The heterogeneity of abnormal patterns and the imbalance between common and rare anomalies further complicate the task. In the context of chest CT, this challenge is amplified by the breadth of abnormalities that may appear, including but not limited to: asbestosis, bronchiectasis, pneumonia, fibrosis, abscess, ground-glass opacities, honeycombing, and various types of nodules (Brixey et al., 2024; Akira et al., 2003; Ma et al., 2022; Baratella et al., 2021; Chaganti et al., 2020; Pu et al., 2021; Gaillandre et al., 2023). The diversity and subtlety of these patterns make it difficult to simulate or synthesize anomalies or collect training data containing various abnormal presentations. These limitations highlight the need for unsupervised frameworks that can learn robust represen-

tations of normal anatomy and identify deviations without relying on human annotations or synthetic anomalies.

Therefore, rather than synthesizing abnormal images which often result in unrealistic or oversimplified pathologies, some works (Zimmerer et al., 2019; Chen et al., 2019; Uzunova et al., 2019; Lu and Xu, 2018; Akcay et al., 2019; Han et al., 2021; Baur et al., 2020; Schlegl et al., 2019; Wolleb et al., 2022; Wyatt et al., 2022; Cai et al., 2025; Bercea et al., 2025) have turned to using generative models to learn the distribution of normal data. In this paradigm, models are trained to capture healthy anatomy, enabling the detection of any out-of-distribution anomalies at inference time as deviations from the learned manifold. This idea has been applied with VAE (Zimmerer et al., 2019; Chen et al., 2019; Uzunova et al., 2019; Lu and Xu, 2018) and GAN-based methods (Akcay et al., 2019; Schlegl et al., 2019; Baur et al., 2020; Han et al., 2021) showing initial success on natural and medical images. However, these approaches are constrained by training instability, mode collapse, and difficulties in capturing high-fidelity structural details (Saad et al., 2024; Kebaili et al., 2023; Sharma et al., 2024). The emergence of diffusion models has introduced a more stable and expressive generative framework. Following the seminal works of denoising diffusion probabilistic models (DDPMs) (Ho et al., 2020; Dhariwal and Nichol, 2021; Song et al., 2020), diffusion-based methods have increasingly been adopted for anomaly detection (He et al., 2024; Beizaee et al., 2025; Wyatt et al., 2022; Zhang et al., 2023; Yao et al., 2025; Yu et al., 2023) for high-fidelity normal image reconstruction. Notably, Wolleb et al. (2022) extended diffusion models to unsupervised anomaly detection in medical contexts. AnoD-DPM (Wyatt et al., 2022) introduces simplex noise during forward diffusion to improve sensitivity in detecting low-frequency anomalies in brain MRI. THOR (Bercea et al., 2024) refines the reverse diffusion process by incorporating implicit guidance via intermediate anomaly maps. These methods (Wolleb et al., 2022; Wyatt et al., 2022; Bercea et al., 2024; Pinaya et al., 2022; Beizaee et al., 2025; Yu et al., 2023) focus on enhancing the quality of generated pseudo-healthy outputs.

Despite aforementioned benchmarks, most anomaly detection methods remain limited in scope, typically focusing on lesion or tumor detection, and fail to exploit anatomical and structural regularities inherent in medical images. Detection accuracy and sensitivity remain areas for improvement. In this work, we propose a diffusion-based framework with scan awareness and consistency enhancement for unsupervised anomaly detection to address these limitations. We summarize our main contributions as follows:

• Awareness of subject anatomy: We integrate a conditioning mechanism to preserve anatomical fidelity in the reconstruction of pseudo-healthy scans, reducing false positives related to incorrect organ shape or boundary.

• Awareness of foreground distinction: We introduce an adaptive masking strategy based on background statistics to binarize anomalies within the organ of interest (lung), accounting for scan-specific variations and heterogeneity of pathological patterns.

• Consistency enhancement and uncertainty awareness via ensembling: We improve detection accuracy and support uncertainty quantification by exploiting the inherent consistency of ensemble inferences from generative models.

• Evaluation on heterogeneous anomalies: We validate on two clinically relevant datasets. The model generalizes across focal and diffuse patterns (nodules, abscesses, fibrosis, ground-glass opacities), demonstrating superior performance compared to existing methods.

## 2. Methodology

### 2.1. Proposed SAC-Diff

The overall workflow of proposed SAC-Diff is illustrated in Fig. 1 and the backbone model is detailed in Fig. 2. Our SAC-Diff is built upon DDPM (Ho et al., 2020), incorporating the simplex noise modifications proposed in Wyatt et al. (2022). We further introduce conditioning and ensembling strategies for scan-aware pseudo-healthy reconstruction and consistency-enhanced detection. These adaptations enable the model to exploit anatomical context and improve robustness.

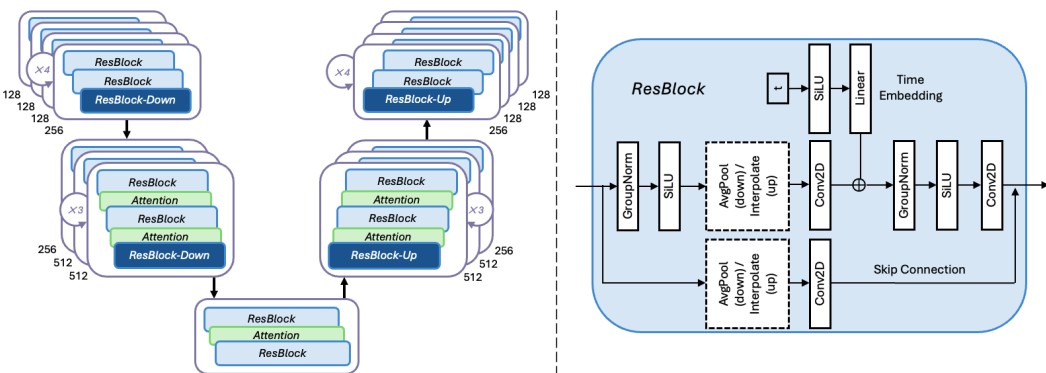

Figure 2: Backbone model of SAC-Diff. Numbers denote feature channels. The network follows a symmetric design with 7 downsampling and 7 upsampling modules (each with 3 residual blocks), connected by a bottleneck module with 2 residual blocks.

### 2.2. Multi-octave Simplex Noise

Gaussian noise used in standard diffusion models has a flat spectral density; however, natural images have been shown to have a power law distribution of frequencies (Ruderman, 1997). Assuming normal and anomalous medical images follow similar power-law characteristics (Metheany et al., 2008; Engstrom et al., 2009; Wyatt et al., 2022), using Gaussian noise can lead to disproportionate corruption. Low-frequency regions, such as large pathological structures, tend to remain relatively uncorrupted during the forward process and are therefore reconstructed in the reverse pass, reducing anomaly detection sensitivity. To address this, simplex noise was introduced for spatially coherent perturbations with stronger low-frequency corruption (Wyatt et al., 2022). Following their approach, we employ multi-octave simplex noise as the forward diffusion corruption, with implementation details provided in Appendix C. The use of multiple octaves is motivated by the fact that the sizes of pathological abnormalities may span a wide range of scales, and combining multiple noise octaves introduces perturbations at both coarse and fine resolutions. This multi-scale corruption enables the diffusion model to effectively perturb and reconstruct abnormalities of varying sizes.

## 2.3. Subject-Anatomy-Aware Conditioning

Standard DDPMs are trained to model the unconditional distribution $p(x_0)$ of the data. In medical imaging applications, anatomical structures such as organs or tissue boundaries are known a priori and can serve as useful conditioning signals to guide the generative process (Guo et al., 2023b, 2024). To incorporate such knowledge, we extend the formulation to a conditional generative model $p_\theta(x_{t-1} \mid x_t, c)$ where $c$ denotes the auxiliary information, in our setting, a binary segmentation mask of the lung field acquired from a lightweight, pre-trained segmentation model (Chaganti et al., 2020). Since only lung slices need to be processed, the segmentation is a necessary step in extracting the lung region and thus does not add overhead or introduce additional computational burden. We adopt an early fusion strategy, where the conditioning signal $c$ is concatenated channel-wise with the noised image $x_t$ as the input to the denoising network at each step, formulated as $\hat{\epsilon}_\theta([x_t \oplus c], t)$. Since the conditioning mechanism is agnostic to disease labels, and the segmentation network has been trained across multiple datasets and disease types in prior work (Chaganti et al., 2020), it demonstrates reasonable robustness to anatomical and pathological variation.

---

**Algorithm 1:** Training

---

$x_0 \sim q(x_0)$
$t \sim \mathcal{U}(\{1, \ldots, T = 1000\})$
$\epsilon \sim \text{Simplex}(\nu = 2^{-6}, N = 6, \gamma = 0.8)$
**repeat**
    Compute noisy input $x_t = \sqrt{\bar{\alpha}_t} \cdot x_0 + \sqrt{1 - \bar{\alpha}_t} \cdot \epsilon$
    Take a gradient descent step on $\nabla_\theta \left\| \epsilon - \hat{\epsilon}_\theta([x_t \parallel c], t) \right\|^2$
**until** *converged*;

---

---

**Algorithm 2:** Sampling (Detection)

---

**Input:** Query image $x$, condition $c$
$\epsilon \sim \text{Simplex}(\nu = 2^{-6}, N = 6, \gamma = 0.8)$
Construct noisy input $x_T = \sqrt{\bar{\alpha}_T}\, x + \sqrt{1 - \bar{\alpha}_T}\, \epsilon$
**for** $t = T_{sampling}, T_{sampling} - 1, \ldots, 1$ **do**
    Predict noise: $\hat{\epsilon}_\theta([x_t \oplus c], t)$
    Compute mean $\mu_\theta(x_t, t, c) = \frac{1}{\sqrt{\alpha_t}} \left( x_t - \frac{1 - \alpha_t}{\sqrt{1 - \bar{\alpha}_t}} \hat{\epsilon}_\theta([x_t \oplus c], t) \right)$
    $z \sim \text{Simplex}(\nu = 2^{-6}, N = 6, \gamma = 0.8)$ if $t > 1$, else $z = 0$
    $x_{t-1} = \mu_\theta(x_t, t, c) + \sigma_t z$
**end**
**return** $\mathcal{E}(x) = \|x - x_0\|^2$

---

## 2.4. Background-Aware Adaptive Masking

We denote $T_{\text{training}}$ as the number of diffusion timesteps used during training (see Alg. 1), and $T_{\text{sampling}}$ as the number of diffusion timesteps used during inference. During inference (see Alg. 2), we apply a partial forward diffusion ($T_{\text{sampling}} < T_{\text{training}}$) to each query sample

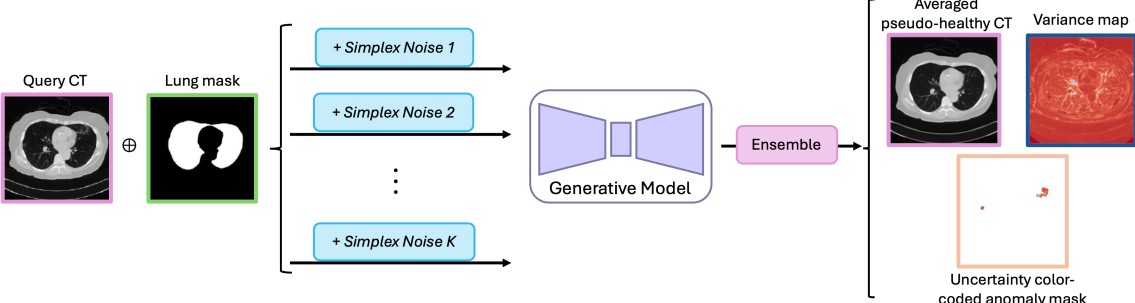

Figure 3: Consistency-enhanced ensembling inference strategy. Each query image is perturbed with $K$ noise realizations. The mean and variance of the resulting reconstructions are used for robust anomaly detection and uncertainty estimation.

from the abnormal datasets. As in current unsupervised diffusion-based AD works (Wang et al., 2024), this ensures that the corrupted image preserves anatomical information, while still introducing enough perturbation to enable effective reconstruction and detection. Following reconstruction, we compute the voxel-wise squared error between the original input $x$ and the reconstructed sample $x_0$. The resulting error map $\mathcal{E}(x) = \|x - x_0\|^2$ serves as an initial estimation of the anomaly map, where larger values indicate deviations from the learned distribution of normal anatomy.

As the intensity range of the generated class activation maps varies significantly across heterogeneous anomalies and scan-specific characteristics (Guo et al., 2023a), fixing a threshold for anomaly mask binarization is suboptimal. To address this, we propose an adjustable background-adaptive thresholding strategy to determine the cutoff used for binarizing anomaly masks. Specifically, we compute a volume-specific threshold $\alpha_a$ based on the statistics of the background region (i.e., voxels outside the lung field): $\alpha_a = \text{mean}(\mathcal{E}[c < 1]) + \lambda \cdot \text{std}(\mathcal{E}[c < 1])$, where $\mathcal{E}$ denotes the predicted MSE map, $c$ is the binary lung mask for the corresponding CT volume, $\mathcal{E}[c < 1]$ denotes $\mathcal{E}$ outside the lung region ($c < 1$), and $\lambda$ is an adjustable parameter.

By changing $\lambda$, we can control how strict or lenient the threshold is for detecting anomalies based on how far a voxel's MSE deviates from the background distribution. The threshold $\alpha_a$ captures both the background bias and the noise level specific to each scan, which yields more robust and consistent segmentation of abnormal regions across subjects with varying intensity distributions. A fully connected conditional random field (CRF) (Krähenbühl and Koltun, 2012) is further applied to enforce spatial consistency, readability and smoothness of the anomaly masks (see Appendix E). The unary potential is derived from per-voxel reconstruction error. Gaussian pairwise potential enforces spatial smoothness across the image via iterations of mean-field inference. As part of the pipeline, this is uniformly applied to all baselines and our methods to ensure a fair comparison. Morphological operations (erosion followed by dilation) are also used to remove noise and close small gaps in the detected regions.

### 2.5. Consistency-Enhanced Ensembling

Figure 3 illustrates our proposed consistency-enhanced ensembling strategy for inference-time detection. Specifically, multiple $(K)$ pseudo-healthy reconstructions are averaged to produce a stable anomaly estimate $\bar{x}_0 = \frac{1}{K} \sum_{k=1}^{K} x_0$, while their voxel-wise standard deviation $\sigma(x) = \left( \frac{1}{K} \sum_{k=1}^{K} (x_0^{(k)} - \bar{x}_0)^2 \right)^{1/2}$ provides an estimate of epistemic prediction uncertainty (He et al., 2025), offering insight into model confidence. The effectiveness of using multiple draws followed by aggregation has been validated in prior generative model–based studies (Whang et al., 2022; Ekmekci and Cetin, 2023). We adopt this concept, use a multi-sample inference strategy within diffusion models for anomaly detection and leverage it to quantify uncertainty in the resulting anomaly masks. In contrast to conventional single-sample inference, our approach exploits the inherent consistency across an ensemble of posterior samples generated by the model. Aggregating these samples improves robustness, mitigates spurious predictions and hallucinations, and enables interpretable voxel-wise uncertainty estimation.

## 3. Experiments

### 3.1. Dataset

Our dataset includes a cohort of chest CT volumes including (a) a total of 252 healthy subjects with no abnormal or actionable findings in the lung, (b) 23 subjects diagnosed with COVID-19, and (c) 23 subjects with ILD. The test datasets present different anomaly characteristics: the ILD cohort contains small, localized, and subtle abnormalities, while the COVID-19 cohort exhibits large, spatially extensive lesions, allowing evaluation across different pathology scales. For the generalization and robustness of the model, both training and testing datasets are heterogeneous, including images acquired from different scanner manufacturers and imaging protocols, reconstructed using varying parameters, containing both contrast-enhanced and non-contrast scans. A list of detailed descriptions of the dataset in Appendix H. All CT scans were reconstructed with sharp reconstruction kernels and calibrated with a slice thickness of 5 mm and an in-plane resolution of $512 \times 512$ pixels. Each volume contains approximately 80 axial slices. Pre-processing included standard CT chest windowing to normalize intensities into [-1, 1]. For all chest CT scans, the lung masks were automatically extracted using a pre-developed lung segmentation model (Chaganti et al., 2020). In addition to conditioning, we also used lung masks to select CT slices that are within the lung region for diffusion model training and anomaly detection inference, yielding a total of 22,734 normal slices. Voxel-wise dense annotations of abnormal regions were provided by thoracic radiologists for all abnormal cases.

### 3.2. Implementation

The SAC-Diff model was trained on a single NVIDIA H100 GPU with a batch size of 4 for 125 epochs. We used the AdamW optimizer (Loshchilov and Hutter, 2018) with an initial learning rate of $1 \times 10^{-4}$ and cosine weight decay. The training objective was the L2 noise-prediction loss, which corresponds to optimizing a simplified variational evidence lower bound. Based on empirical experimental results, we set $T_{\text{training}} = 1000$, $T_{\text{sampling}} = 550$,

$\lambda = 1$ for COVID dataset and $\lambda = -0.25$ for ILD dataset, and number of samples $K = 7$ in ensemble. Input data consisted of 2D axial slices, which were stacked to obtain 3D outputs. To ensure slice consistency within each subject, a fixed random seed is used at inference time to generate the same initial simplex noise for each 3D volume, ensuring a consistent noise pattern across slices that stabilizes reconstruction.

The trained model was evaluated using dense annotations on both COVID-19 and ILD datasets, each containing 20 subjects. Qualitatively, we compare the reconstructed pseudo-healthy images against their original input CT scans, and the anomaly maps along with segmentation results against the radiologist-annotated ground truth. Quantitatively, we report mean Intersection over Union (IoU), Dice Similarity Coefficient (Dice), Precision, Recall, and False Positive Rate (FPR) on both test sets.

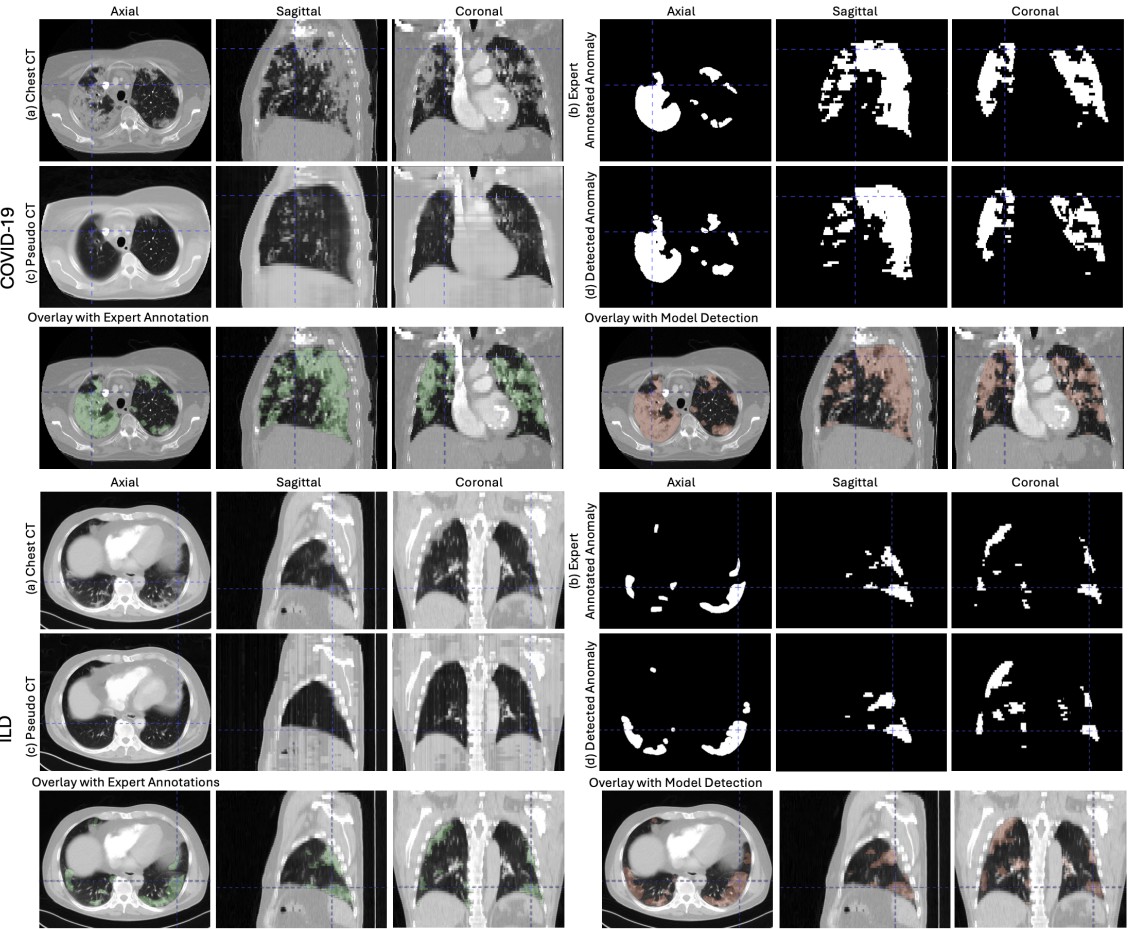

Figure 4: Examples of automatic anomaly detection using the proposed method. (a) 3D chest CT scan from a COVID-19 patient and an ILD patient; (b) Manually annotated anomaly map by radiologist; (c) Synthetic CT scan generated by the model, with anomalies suppressed; (d) Anomaly detected by our model.

### 3.3. Detection Result Visualization

Fig. 4 presents two 3D examples of anomaly detection using our proposed method, one from a COVID-19 case and one from an ILD case. The reconstructed images suppress heterogeneous anomalies while preserving lung structures in both cases. As illustrated across the coronal, sagittal, and axial views, the model effectively captures the size, structure, and location of various anomalies for different diseases. In Appendix G, we show a UMAP of features extracted from healthy, unhealthy, and model-reconstructed pseudo-healthy samples using a foundation model (Pai et al., 2024). Feature-space distances were further computed, yielding 1.1371 between healthy and unhealthy samples and 1.1135 between healthy and reconstructed pseudo-healthy samples.

### 3.4. Baseline Comparison and Ablation Study

We conducted sensitivity tests and ablation studies by comparing SAC-Diff with other baselines and configurations, including a supervised method dedicated to COVID lesions (Biondi et al., 2021), a state-of-the-art medical foundation model MedSAM2 (Ma et al., 2025), and generative model baselines DDPM (Ho et al., 2020) and AnoDDPM (Wyatt et al., 2022). We followed the same training pipeline on the same dataset and optimized the hyperparameter settings used in subsequent experiments.

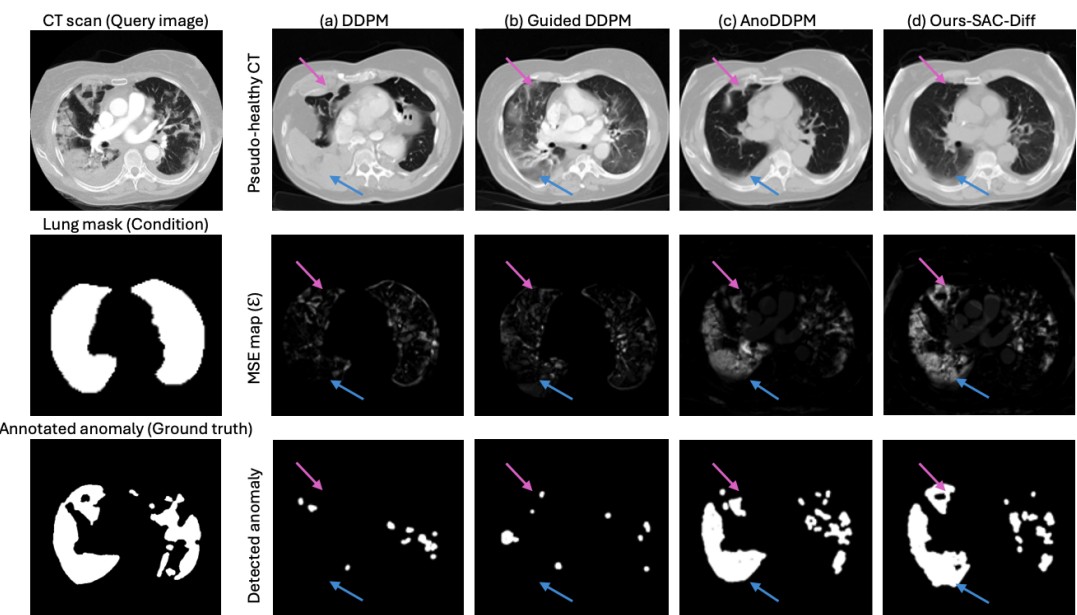

Figure 5: Comparison of anomaly detection between previous models and our proposed model. (a) DDPM; (b) guided DDPM; (c) AnoDDPM; (d) Ours: SAC-Diff. The arrows indicate abnormal regions that were missed by previous methods but successfully detected by the proposed approach.

Qualitative comparisons between the baselines and our model are shown in Fig. 5. In pseudo-normal CT reconstruction, our proposed architecture effectively removes both large

Table 1: Baseline comparison and ablation study on COVID dataset. Checkmarks indicate which components are enabled in each variant. The best results are marked **in bold**. An asterisk (*) denotes statistically significant improvement over prior methods.

| Config | Simplex Noise | Anatomy-Guidance | Background-Adaptive | Ensembling | IoU ↑ | Dice ↑ | Precision ↑ | Recall ↑ | FPR ↓ |
|---|---|---|---|---|---|---|---|---|---|
| COVIDSeg (**Supervised**) (Biondi et al., 2021) | | | | | 0.2088 (0.1672) | 0.3169 (0.2088) | 0.2440 (0.2100) | **0.7033** (0.3230) | 0.0166 (0.0186) |
| MedSAM2 (**Foundation**) (Ma et al., 2025) | | | | | 0.3731 (0.1800) | 0.5186 (0.2014) | **0.5306** (0.2398) | 0.6215 (0.2724) | **0.0076** (0.0089) |
| DDPM (Ho et al., 2020) | | | | | 0.1260 (0.1025) | 0.2099 (0.1560) | 0.1462 (0.1238) | 0.5330 (0.2256) | 0.0228 (0.0199) |
| AnoDDPM (Wyatt et al., 2022) | ✓ | | | | 0.1596 (0.1535) | 0.2487 (0.2046) | 0.2269 (0.1990) | 0.4660 (0.2870) | 0.0215 (0.0181) |
| (a) DDPM Variant | | ✓ | | | 0.1411 (0.1036) | 0.2335 (0.1548) | 0.1715 (0.1322) | 0.4980 (0.2088) | 0.0227 (0.0191) |
| (b) DDPM Variant | | | ✓ | | 0.0230 (0.0251) | 0.0438 (0.0468) | 0.0240 (0.0266) | 0.3551 (0.2776) | 0.0195 (0.0190) |
| (c) AnoDDPM Variant | ✓ | ✓ | | | 0.2147 (0.1943) | 0.3161 (0.2421) | 0.2735 (0.2391) | 0.5092 (0.2705) | 0.0134 (0.0128) |
| (d) SAC-Diff Variant | ✓ | | ✓ | | 0.2635 (0.1749) | 0.3869 (0.2197) | 0.3250 (0.2291) | 0.6204 (0.2300) | 0.0129 (0.0125) |
| (e) SAC-Diff Variant | | ✓ | ✓ | | 0.0189 (0.0258) | 0.0358 (0.0476) | 0.0193 (0.0266) | 0.5025 (0.3653) | 0.0197 (0.0194) |
| SAC-Diff w/o Ensemble | ✓ | ✓ | ✓ | | 0.3140 (0.1939) | 0.4439 (0.2323) | 0.3737 (0.2434) | 0.6849 (0.2404) | 0.0117 (0.0121) |
| SAC-Diff | ✓ | ✓ | ✓ | ✓ | **0.3871** (0.1727)* | **0.5341** (0.1957)* | 0.5008 (0.2483) | 0.6662 (0.2262) | 0.0094 (0.0098) |

Table 2: Baseline comparison and ablation study on ILD dataset.

| Config | Simplex Noise | Anatomy-Guidance | Background-Adaptive | Ensembling | IoU ↑ | Dice ↑ | Precision ↑ | Recall ↑ | FPR ↓ |
|---|---|---|---|---|---|---|---|---|---|
| COVIDSeg (**Supervised**) (Biondi et al., 2021) | | | | | 0.1785(0.1537) | 0.2765(0.2137) | 0.2005(0.1792) | **0.7981**(0.2781) | 0.0282(0.0214) |
| MedSAM2 (**Foundation**) (Ma et al., 2025) | | | | | 0.0270 (0.0350) | 0.0540 (0.0637) | 0.3369 (0.4977) | 0.3324 (0.4626) | **0.0142** (0.0120) |
| DDPM (Ho et al., 2020) | | | | | 0.1193 (0.0991) | 0.2000 (0.1509) | 0.1379 (0.1201) | 0.5384 (0.2274) | 0.0232 (0.0198) |
| AnoDDPM (Wyatt et al., 2022) | ✓ | | | | 0.1535 (0.1467) | 0.2417 (0.1968) | 0.2195 (0.1998) | 0.4613 (0.2769) | 0.0207 (0.0176) |
| (a) DDPM Variant | | ✓ | | | 0.1399 (0.1029) | 0.2319 (0.1529) | 0.1704 (0.1321) | 0.5035 (0.2045) | 0.0230 (0.0189) |
| (b) DDPM Variant | | | ✓ | | 0.0147 (0.0125) | 0.0287 (0.0239) | 0.0151 (0.0129) | 0.4687 (0.2054) | 0.0331 (0.0213) |
| (c) AnoDDPM Variant | ✓ | ✓ | | | 0.1282 (0.1317) | 0.2064 (0.1883) | 0.1516 (0.1624) | 0.5183 (0.2933) | 0.0245 (0.0181) |
| (d) SAC-Diff Variant | ✓ | | ✓ | | 0.1285 (0.1210) | 0.2094 (0.1722) | 0.1499 (0.1580) | 0.6020 (0.2601) | 0.0285 (0.0185) |
| (e) SAC-Diff Variant | | ✓ | ✓ | | 0.0116 (0.0172) | 0.0224 (0.0325) | 0.0117 (0.0173) | 0.5709 (0.3360) | 0.0332 (0.0215) |
| SAC-Diff w/o Ensemble | ✓ | ✓ | ✓ | | 0.2733 (0.1929) | 0.3974 (0.2233) | 0.3954 (0.2392) | 0.5488 (0.2708) | 0.0207 (0.0165) |
| **SAC-Diff** | ✓ | ✓ | ✓ | ✓ | **0.3114** (0.2000)* | **0.4435** (0.2184)* | **0.4921** (0.2405)* | 0.5229 (0.2648) | 0.0179 (0.0156) |

and subtle anomalies while preserving lung boundaries. From visual comparisons, prior methods either fail to preserve structural details or fail to suppress significant anomalies. As highlighted by the pink and blue arrows, SAC-Diff successfully detects abnormal regions missed by previous approaches.

Tables 1 and 2 compare SAC-Diff with prior methods on COVID-19 and ILD datasets, respectively. Compared with other baselines and variants (a–e), the proposed SAC-Diff significantly outperforms all methods on both datasets in IoU and Dice. We include MedSAM2 and COVIDSeg to represent current foundation model–based and supervised baselines relevant to our task. The supervised COVIDSeg achieved the highest recall, but generalized poorly on our in-house COVID set and a more diverse ILD set. MedSAM2, included as a strong recent foundation model baseline with extensive pretraining, exhibited the lowest FPR on both datasets. Despite its on-par performance on the COVID set, both IoU and Dice scores of MedSAM2 dropped drastically on the ILD dataset with more diverse anomalies. The failures of current supervised and foundation models largely stem from the reliance on labeled data and sensitivity to domain changes. These observations highlight such limitations, including limited generalization to unseen or heterogeneous pathologies and sensitivity to domain shifts arising from differences in scanners and acquisition protocols.

Comparing DDPM with variant (a) and AnoDDPM with variant (c), we observe anatomy guidance alone could improve performance for the COVID-19 dataset but was not sufficient for ILD. Simplex noise yields consistent improvements across settings, as seen in (a) vs. (c) and (b) vs. (d). When background-adaptive thresholding is combined with simplex noise, it achieves strong performance, as in variant (d). In addition, ensembling further improves the model's anomaly detection performance through the enhanced inter-sample consistency.

With all the components, the proposed SAC-Diff significantly outperforms other baselines in IoU and Dice, achieving an average IoU of 0.3871 (+3.75%) and an average Dice score of 0.5341 (+2.99%) on COVID-19, and average IoU of 0.3114 (+74.45%) and an average Dice score of 0.4435 (+60.40%) on ILD.

In addition to improved accuracy, the ensembling strategy enables uncertainty quantification for the model's predictions by computing the voxel-wise standard deviation across reconstructed results. The resulting uncertainty map provides a per-voxel measure of confidence, offering insights into the model's reliability and facilitating a more robust clinical interpretation. We illustrate this effect with three representative examples in Fig. 6. The predicted anomaly maps are visualized using a color scale: warmer colors indicate higher model confidence related to high cross-inference consistency, while cooler colors reflect greater uncertainty. The arrows highlight regions where ensembling leads to improved detection.

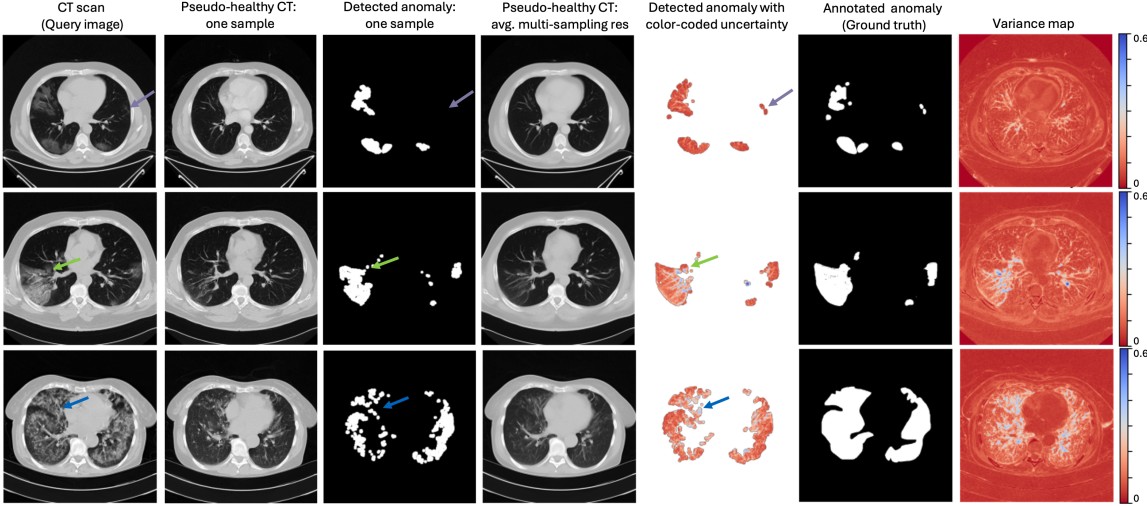

Figure 6: Examples of automatic anomaly detection results with ensembling enabled. The predicted anomaly maps and variance maps are represented using the same color scale, where warmer colors correspond to higher model confidence. The arrows highlight regions where inherent consistency in ensembling leads to improved detection.

## 4. Discussion and Conclusion

In this work, we presented SAC-Diff for unsupervised anomaly detection in chest CT. SAC-Diff uses simplex noise perturbation, subject-aware anatomical conditioning, background-aware masking, and ensemble inference with uncertainty quantification to deliver reliable anomaly detection. Experiments show that SAC-Diff consistently outperforms previous supervised, diffusion-based, and foundation model-based methods, highlighting its ability to localize heterogeneous anomalies. We summarize key discussion points and outline potential directions for future work as follows:

**Domain Shift.** Our training dataset contains primarily non-contrast CT images, whereas the test sets include contrast-enhanced scans. This domain shift may slightly affect performance, as contrast agents alter intensity distributions, though mostly outside the lung (see Fig. 4 COVID-19 (a) and (c)). While the background awareness of SAC-Diff shows strong robustness under this shift, future work should address contrast variability through domain adaptation techniques or include contrast-enhanced scans into training.

**Model Interpretability.** The ensembling improves reliability and interpretability through uncertainty quantification. On the uncertainty maps, low variance indicates high model confidence achieved under inherent consistency, and high variance indicates detection with potential ambiguity. This reduces the risk of over-reliance on spurious predictions, making it more suitable for integration into real-world clinical workflows.

**Dependence on Segmentation Accuracy.** Since the model is guided by a lung mask, the segmentation accuracy impacts anomaly detection. Anomalies along the chest wall or airway borders, such as wall thickening, are therefore relatively difficult to capture. This limitation underscores the need for task-specific lung field segmentation methods tailored for anomaly detection. In this work, we used a pre-trained network for lung segmentation (Chaganti et al., 2020); however, developing an end-to-end pipeline that jointly optimizes organ segmentation and anomaly detection can be a promising direction.

**Conditioning Strategies.** To provide subject-aware anatomical guidance, we condition the model on a lung mask, which informs the denoising process at every step. This simple yet effective conditioning improves anomaly localization using anatomical context. Beyond lung masks, conditioning on positional encoding or adjacent CT slices could further preserve lung boundaries and enforce spatial coherence across volumes. In our setting, the lung mask serves as a binary coarse anatomical prior and our goal is to use the lung mask to restrict the model to anatomically plausible regions. While early fusion is empirically efficient and stable in our case, more expressive conditioning mechanisms, such as cross-attention (Rombach et al., 2022), SPADE (Park et al., 2019), or FiLM-based feature modulation (Perez et al., 2018), can enable richer interactions between $x_t$ and $c$. As shown in Appendix D, given the low-dimensional and geometric nature of our conditioning signal, early fusion via channel concatenation is empirically sufficient for our task. We leave a more comprehensive investigation of alternative conditioning strategies for future work.

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

## Appendix A. Lung Mask Segmentation

For lung mask segmentation (Chaganti et al., 2020), the network was first trained on large-scale datasets (8087 patients, 8006 for training and 81 for validation), and then fine-tuned on 1136 patients (trained on 1076 and validated on 60) with multiple abnormality patterns, including ground-glass opacities, consolidation, effusions, masses, and others to ensure robustness in abnormal scans.

Multiscale deep reinforcement learning was used to identify anatomical landmarks (Ghesu et al., 2019). The carina bifurcation serves as the primary reference point for locating the lung region of interest (ROI); when this landmark cannot be detected, the sternum tip is used as an alternative. The dimensions and spatial placement of the lung ROI relative to the detected landmark are defined based on annotated lung datasets.

The extracted lung ROI is then resampled to a 2-mm isotropic grid and passed through an adversarial Image-to-Image Network (DI2IN) (Yang et al., 2017) to produce the lung segmentation. The main structure of DI2IN is a symmetric convolutional encoder-decoder and the discriminator is a CNN. Afterward, the predicted ROI segmentation is mapped back to the original image space to restore the native resolution and dimensions. This pipeline is pre-trained on a heterogeneous patient population to ensure performance across diverse pathologies and applied to our datasets.

## Appendix B. Denoising Diffusion Probabilistic Models

DDPMs (Ho et al., 2020; Nichol and Dhariwal, 2020) have emerged as a state-of-the-art approach in generative modeling, achieving high sample fidelity and superior mode coverage. It consists of two core components:

**Forward diffusion process:** The forward process $q(x_t|x_{t-1})$ gradually corrupts a clean sample $x_0 \sim q(x_0)$ into Gaussian noise over a sequence of $T$ time steps. At each step, the $x_t$ is sampled from a Gaussian distribution centered around the previous state:

$$q(x_t|x_{t-1}) = \mathcal{N}(x_t; \sqrt{\alpha_t}x_{t-1}, (1 - \alpha_t)\mathbf{I}),$$

where $\{\alpha_t\}_{t=1}^T$ is a predefined noise schedule. By recursively applying this process, a closed-form expression for the noised input at any timestep $t$ is

$$x_t = \sqrt{\bar{\alpha}_t}x_0 + \sqrt{1 - \bar{\alpha}_t}\,\epsilon,$$

where $\bar{\alpha}_t = \prod_{s=1}^t \alpha_s$ and $\epsilon \sim \mathcal{N}(0, \mathbf{I})$.

**Learned reverse denoising process:** The generative process in DDPMs is defined by a learned reverse Markov chain:

$$p_\theta(x_{0:T}) = p(x_T)\prod_{t=1}^T p_\theta(x_{t-1}|x_t),$$

where the prior is defined as $p(x_T) = \mathcal{N}(0, \mathbf{I})$. Each reverse step is modeled as a Gaussian distribution with a mean $\mu_\theta(x_t, t)$ and fixed variance. The mean can be reparameterized using a neural network $\hat{\epsilon}_\theta(x_t, t)$ that predicts the noise used in the forward process

$$\mu_\theta(x_t, t) = \frac{1}{\sqrt{\alpha_t}}\left(x_t - \frac{1 - \alpha_t}{\sqrt{1 - \bar{\alpha}_t}}\hat{\epsilon}_\theta(x_t, t)\right).$$

This parameterization allows us to train the model to predict the noise $\epsilon \sim \mathcal{N}(0, \mathbf{I})$ that perturbed $x_0$ into $x_t$. The model is optimized using a denoising score-matching objective:

$$\mathcal{L}(\theta) = \sum_{t=1}^{T} \mathbb{E}_{x_0 \sim q(x_0), \epsilon \sim \mathcal{N}(0,\mathbf{I})} \left[ \|\epsilon - \hat{\epsilon}_\theta(x_t, t)\|_2^2 \right].$$

We adapted the DDPM framework described above, replacing the Gaussian noise with Simplex noise.

## Appendix C. Generation of Simplex Noise

We followed the work of Wyatt et al. (2022) for generating multi-octave simplex noise: Perlin noise is generated by sampling random gradients on a regular lattice. The inner products between the gradients and the offsets from the nearest 4 lattice points are computed and interpolated. Simplex noise replaces lattice with a simplex grid of equilateral triangles. Compared with Gaussian noise, simplex noise introduces more coherent and structured corruptions. Rather than using a single-scale simplex noise function, we used a number of octaves of noise by combining $N$ frequencies of noise together, where the next frequency's amplitude reduces by $\gamma$. Using a base frequency $\nu = 2^{-6}$, multi-octave simplex noise with $N = 6$ and $\gamma = 0.8$ was constructed.

## Appendix D. Conditioning Mechanism

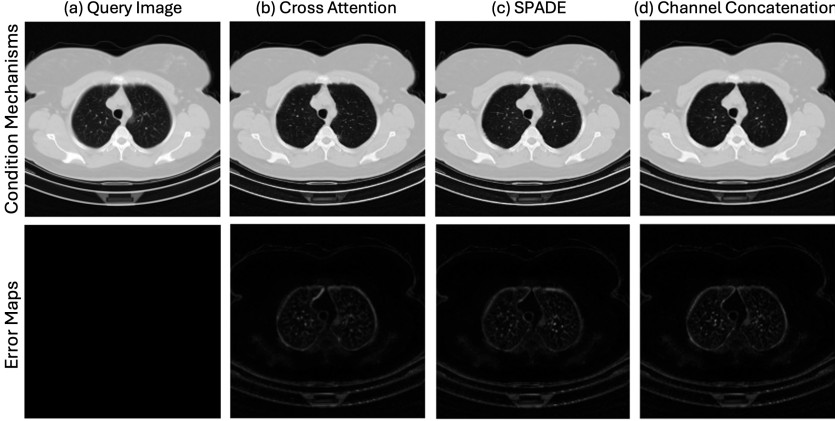

Figure A1: Qualitative comparison of conditioning mechanisms using representative healthy CT scan reconstructions (Guo et al., 2025): (a) query image, (b) cross-attention, (c) SPADE, and (d) channel concatenation, along with their corresponding error maps.

## Appendix E. Effect of Conditional Random Field Post-Processing

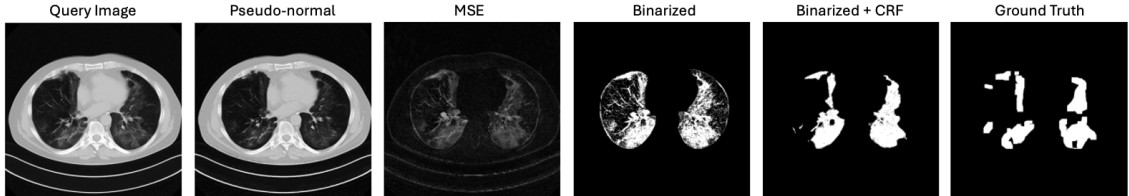

Figure A2: Sample results from initial experiments applying dense CRF post-processing to MSE maps. Empirically, we observe that the CRF algorithm improves the spatial consistency and smoothness of the resulting anomaly masks.

## Appendix F. Effect of $T_{\text{sampling}}$ on Model Performance

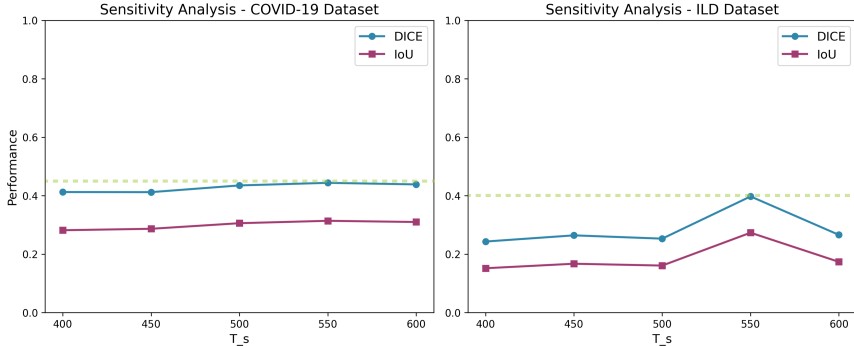

Figure A3: Sensitivity curves showing the effect of sampling timesteps ($T_{\text{sampling}}$) on model's anomaly detection performance. Dice and IoU scores illustrate how AD quality varies with $T_{\text{sampling}}$.

## Appendix G. UMAP Using Extracted Embeddings

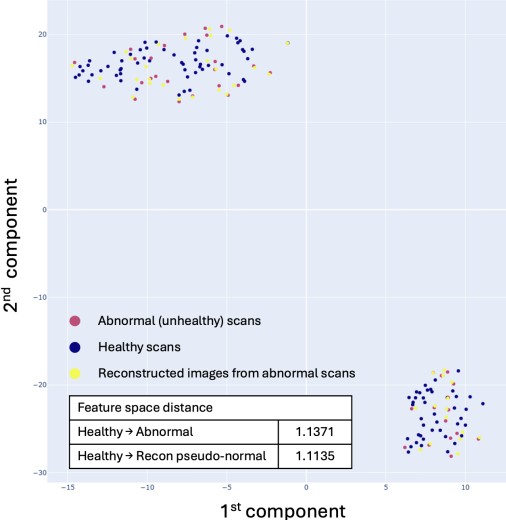

Figure A4: UMAP visualization of features extracted from healthy normal scans, abnormal scans, and reconstructed scans. While we did not observe separated clusters in the embedding space, the plot indicates that reconstructed images do not form a distinct manifold.

## Appendix H. Additional Dataset Information

Table A1: Datasets categorized by manufacturer model

| Dataset | Manufacturer Model | Count |
|---|---|---|
| Training dataset | Siemens Emotion 16 | 56 |
| | Siemens Sensation Cardiac 64 | 47 |
| | Siemens SOMATOM Scope | 46 |
| | Siemens Emotion 16 (2010) | 44 |
| | Siemens Biograph 16 | 31 |
| | Siemens SOMATOM Definition AS | 15 |
| | Siemens Sensation 16 | 10 |
| | Siemens Emotion 6 | 3 |
| Testing dataset – COVID | Siemens Sensation 64 | 8 |
| | Siemens SOMATOM go.Up | 3 |
| | Philips Ingenuity CT | 3 |
| | Canon Aquilion ONE | 2 |
| | Siemens SOMATOM Edge Plus | 2 |
| | Siemens SOMATOM X.cite | 1 |
| | Unknown | 1 |
| Testing dataset – ILD | Siemens SOMATOM Definition AS | 15 |
| | Siemens SOMATOM Force | 2 |
| | Siemens SOMATOM Definition AS+ | 1 |
| | Siemens Biograph 128 | 1 |
| | Siemens SOMATOM Volume Zoom | 1 |

Table A2: Distribution of reconstruction kernels across datasets

| Dataset | Reconstruction Kernel | Count |
|---|---|---|
| Training dataset | B70s | 155 |
| | B70f | 95 |
| | I70f 3 | 2 |
| Testing dataset – COVID | B60f | 7 |
| | Br60f 3 | 3 |
| | YB | 3 |
| | FC55 | 2 |
| | Br64f 2 | 1 |
| | FC05 | 1 |
| | Br59f 3 | 1 |
| | Bl57f 3 | 1 |
| | Unknown | 1 |
| Testing dataset – ILD | B70f | 12 |
| | I70f 3 | 4 |
| | Bv59d 3 | 1 |
| | I70f 2 | 1 |
| | Bl57d 3 | 1 |
| | B70s | 1 |

Table A3: Distribution of Kilovoltage Peak across datasets

| Dataset | Kilovoltage Peak (kVp) | Count |
|---|---|---|
| Training dataset | 130 | 148 |
| | 120 | 100 |
| | 100 | 3 |
| | 110 | 1 |
| Testing dataset – COVID | 120 | 12 |
| | 110 | 4 |
| | 100 | 3 |
| | Unknown | 1 |
| Testing dataset – ILD | 120 | 11 |
| | 100 | 6 |
| | 80 | 2 |
| | 140 | 1 |

Table A4: Distribution of protocols across datasets

| Dataset | Protocol | Count |
|---|---|---|
| Training dataset | Thorax Native | 171 |
| | Chest with Lower Extremities | 32 |
| | Pulmonalis | 17 |
| | Chest Routine Non-Contrast | 16 |
| | Thorax Low Dose | 16 |
| Testing dataset – COVID | Torax | 7 |
| | DeIdentified | 4 |
| | TX1-TX2 TORAX | 3 |
| | TX1-TORAX S/C / Torax | 3 |
| | TÓRAX | 1 |
| | Chest wo Virtual Bronch | 1 |
| | Unknown | 1 |
| Testing dataset – ILD | Thorax Native | 9 |
| | Thorax KM | 3 |
| | Thorax N | 1 |
| | FDG PETCT Wholebody | 1 |
| | Thorax KMHR | 1 |
| | Thorax Abdomen | 1 |
| | BWS LWS | 1 |
| | Thorax Native S | 1 |
| | Thorax Routine | 1 |
| | Unknown | 1 |

