# OpenReview forum: "SAC-Diff: A Scan-Aware Consistency-Enhanced Diffusion Framework for Unsupervised Chest CT Anomaly Detection"
_MIDL.io/2026/Conference — MIDL 2026 Poster_

### Official Review · Reviewer_iTgv · 2026-01-07

**Confidence:** 3
**Preliminary Rating:** 2
**Final Rating:** 2

**Summary:**

This work is about unsupervised abnormality detection for lung disease screening from CT scans. Authors addresses the challenges of 1) hallucination of unsupervised generative models because of the gaussian design, and 2) insufficient anatomical guidance. The proposed method is SAC-Diff, a diffusion-based framework with the integration of A) simplex noise, B) subject-aware anatomical priors and C) background-aware masking, for robust modelling of scan-specific variations and heterogeneous lung abnormalities. Models are trained and evaluation on two datasets including various abnormalities.

**Strengths:**

1. The paper presents a compelling motivation, clearly articulating both technical and clinical challenges in medical imaging, including anatomical variability, heterogeneous and rare abnormalities, and the high cost of human annotations.

2. The proposed SAC-Diff framework is rigorously evaluated on two datasets encompassing diverse abnormality types, which strengthens the empirical credibility of the approach. The experimental validation includes an ablation study and qualitative analysis, providing multiple perspectives on model behaviours. Additionally, the evaluation protocol (train/val/test splits) is mentioned.

3. The authors demonstrate commendable transparency regarding limitations, including reliance on accurate lung segmentation and potential domain shifts between non-contrast and contrast-enhanced CT scans.

4. To address suboptimality in anomaly-mask binarization, the authors propose a background-adaptive thresholding strategy. This method enhances robustness across abnormalities with varying visual characteristics, and ablation results suggest that it has one of the highest significant impacts on performance.

**Weaknesses:**

1. The method extends a standard DDPM framework with components such as simplex noise, lung-mask conditioning, and ensemble-based inference. While effective, these extensions, particularly ensemble-based inference, may be perceived as incremental performance boosts rather than principled modelling contributions, limiting the methodological novelty.

2. The manuscript lacks formalism, with few equations or explicit definitions, which hampers accessibility and makes it harder for readers to fully understand the model on first reading.

3. Some assumptions and claims are insufficiently supported, both empirically and bibliographically. For example, the discussion of power-law behaviour in Section 2.2.

4. Although the reliance on accurate lung segmentation is acknowledged, it may also limit generalizability to other anatomical regions or imaging modalities, such as abdominal CT.

**Detailed Comments:**

1. Quantitative results: Improve clarity by separating main comparisons from ablation studies, either into two tables or clearly distinct sections. Place the supervised baseline first to distinguish it from unsupervised methods. For the ablation study, focus on key metrics (e.g., IoU and Dice) and include a dedicated table showing DDPM, its variants, and the final SAC-Diff model. Group main results per dataset for readability.

2. Qualitative results: Eventually present input slices with predicted masks overlaid to clearly illustrate localization and model behaviour.

**Justification Of Final Rating:**

I thank the authors for their response and for the revisions made to the manuscript.

This work addresses unsupervised abnormality detection from CT scans through a DDPM-like framework that integrates: (i) a lung-aware conditioning strategy, (ii) an adaptive masking mechanism to guide abnormality prediction based on background and organ-specific information, and (iii) an ensembling strategy to enhance the final predictions.

While the method is evaluated on two datasets covering a range of abnormalities and demonstrates strong empirical performance supported by ablation studies, the core concerns regarding reproducibility and methodological rigor remain. In particular, the reliance on internal datasets, the absence of code release, and the lack of precise methodological formalization make it difficult to fully assess or reproduce the proposed approach.

**Justification Of The Preliminary Rating:**

The paper tackles an important problem in medical imaging: the development of robust and interpretable unsupervised methods for abnormality detection. The proposed method extends DDPM with incremental contributions, including anatomy-aware priors, multi-octave simplex noise to better corrupt low-frequency regions, background-adaptive prediction, and ensemble-based inference. While the empirical results are promising, the work would be strengthened by: 1) A more comprehensive bibliography or empirical evidence to support key assumptions in the motivation; 2) A more rigorous and precise methodological formalization; and 3) Clearer presentation of quantitative results to highlight the contribution of each component. Additionally, the absence of source code release and the use of internal datasets may limit reproducibility.

**Questions To Address In The Rebuttal:**

1. Introduction: The paper states that autoencoder- and GAN-based methods struggle to capture high-fidelity structural details. Could the authors provide supporting references or empirical evidence for this claim?

2. Methology (2.2): The assumption that normal and anomalous medical images share similar power-law characteristics is made. What prior work or empirical observations support this assumption?

3. Methodology (2.4): Could the authors clarify the definitions and roles of T_training and T_sampling ?

4. Baselines: What is the rationale for including MedSAM2 as a baseline? Given its heavy pretraining, how do the authors justify the fairness of comparing it with SAC-Diff, especially since SAC-Diff requires ensemble inference to outperform it, whereas it already surpasses other supervised and unsupervised baselines without ensembling?

5. Methology (Section 2.2): The paper proposes multi-octave simplex noise to address the issue of large pathological structures remaining uncorrupted during diffusion. Have the authors analyzed whether performance gains correlate with pathology size?

6. Datasets: Are the datasets used internal, external, or a mix?

---

> ### Author Response · Authors · 2026-01-25
> **Response to Reviewer iTgv**
>
> We would like to thank the reviewer for taking the time to evaluate our paper and for providing valuable feedback. We are pleased to address the questions to respond to the reviewer’s thoughtful comments, including the model’s assumption, baseline selection and dataset. These updates have been incorporated into the revised manuscript to further strengthen the paper.
>
> **(1) Introduction: The paper states that autoencoder- and GAN-based methods struggle to capture high-fidelity structural details. Could the authors provide supporting references or empirical evidence for this claim?**
>
> Thank you very much for the feedback. We revised the manuscript to **include supporting references** demonstrating the limitations of these methods in the Introduction section: “However, these approaches are constrained by training instability, mode collapse, and difficulties in capturing high-fidelity structural details **(Saad et al., 2024; Kebaili et al., 2023; Sharma et al., 2024)**.”
>
> **(2) Methology (2.2): The assumption that normal and anomalous medical images share similar power-law characteristics is made. What prior work or empirical observations support this assumption?**
>
> We appreciate the reviewer’s insightful question regarding the model assumption. Prior work has shown that natural image power spectra tend to decay as a power law (Ruderman, 1997). In medical imaging, there is no direct evidence in the literature that normal and anomalous images share identical power-law characteristics. However, several studies (Metheany et al., 2008; Engstrom et al., 2009) demonstrate that **CT and mammography images exhibit approximate power-law behavior**, even though these works do not explicitly compare normal and pathological images. Finally, this power-law assumption has been **adopted in prior anomaly detection work** (Wyatt et al., 2022), where the authors empirically show that low-frequency-dominated, structured noise improves anomaly reconstruction performance. We have added these references to support the assumption in the revised manuscript.
>
> **(3) Methodology (2.4): Could the authors clarify the definitions and roles of T_training and T_sampling ?**
>
> We thank the reviewer for the question. T_training  denotes the number of diffusion timesteps used during training, while T_sampling refers to the number of diffusion timesteps used during inference. We have **added these definitions** to the revised manuscript in **Section 2.4** for clarity: “We denote T_training as the number of diffusion timesteps used during training, and T_sampling as the number of diffusion timesteps used during inference.”
>
> **(4) Baselines: What is the rationale for including MedSAM2 as a baseline? Given its heavy pretraining, how do the authors justify the fairness of comparing it with SAC-Diff, especially since SAC-Diff requires ensemble inference to outperform it, whereas it already surpasses other supervised and unsupervised baselines without ensembling?**
>
> We appreciate the reviewer’s insightful comment. MedSAM2 is included as **a strong recent foundation model baseline with extensive pretraining**. While it outperforms SAC-Diff without ensemble on COVID dataset, it demonstrates limited generalization to ILD dataset where it is outperformed by SAC-Diff without ensembling. Notably, SAC-Diff **already surpasses all other baselines in terms of IoU and Dice without ensembling** and further improves to outperform all baselines when ensemble inference is applied. We include MedSAM2 and COVIDSeg to **represent current foundation model-based** and **supervised baselines within the context of our task**. This rationale has been clarified in the revised manuscript:
>
> “We include MedSAM2 and COVIDSeg to represent current foundation model-based and supervised baselines relevant to our task. The supervised COVIDSeg achieved the highest recall, but generalized poorly on our in-house COVID set and a more diverse ILD set. MedSAM2, included as a strong recent foundation model baseline with extensive pretraining, …”

---

> > ### Author Response · Authors · 2026-01-25
> > **Response to Reviewer iTgv**
> >
> > **(5) Methology (Section 2.2): The paper proposes multi-octave simplex noise to address the issue of large pathological structures remaining uncorrupted during diffusion. Have the authors analyzed whether performance gains correlate with pathology size?**
> >
> > Thank you very much for raising this insightful observation. The motivation for using multi-octave noise is that **pathological abnormalities can vary substantially in size**, and injecting noise at multiple scales allows the diffusion process to **perturb both small and large structures**. While we did not explicitly stratified performance by pathology size, in our study this aspect is implicitly evaluated through the use of different datasets with markedly different anomaly characteristics: the ILD dataset predominantly contains **small, localized, and subtle abnormalities**, whereas the COVID-19 dataset contains a lot of **large and consolidated** lesions. Our studies show consistent performance improvements from multi-octave simplex noise on both datasets compared to baselines, indicating the noise design is effective across different pathology sizes.
> >
> > We updated **section 2.2** to include: “The use of multiple octaves is motivated by the fact that the size of pathological abnormalities may span a wide range of scales, and combining multiple noise octaves introduces perturbations at both coarse and fine resolutions. This multi-scale corruption enables the diffusion model to effectively perturb and reconstruct abnormalities of varying sizes.”
> >
> > We also updated **section 3.1**: “The test datasets present different anomaly characteristics: the ILD cohort contains small, localized, and subtle abnormalities, while the COVID-19 cohort exhibits large, spatially extensive lesions, allowing evaluation across different pathology scales.”
> >
> > We hope this clarifies the point and appreciate the reviewer’s attention.
> >
> > **(6) Datasets: Are the datasets used internal, external, or a mix?**
> >
> > Thank you very much for raising this question. All datasets used in this study are internal datasets collected from our clinical partners. While external validation is desirable, we note that **our experiments span multiple pathologies, scanner variations, different imaging protocols, different reconstruction parameters, and include both contrast-enhanced and non-contrast scans**. To address concerns regarding generalizability, we provide detailed descriptions of the datasets in **Appendix G**.
> >
> > **(7) Presentation Format Improvement**
> >
> > Finally, for **improving the presentation of quantitative results**, we have reordered the baselines so that supervised and foundation models appear first, and we have clearly labeled the variants in the table for improved readability. We kept the baseline comparisons and ablation study in a single table to avoid presenting duplicate results of the final SAC-Diff model. For **the qualitative results**, we have added an additional row showing input slices overlaid with predictions, to more clearly illustrate the localization of the detected anomalies. Please see the revised manuscript.

---

### Official Review · Reviewer_hpuJ · 2026-01-09

**Confidence:** 5
**Preliminary Rating:** 3
**Final Rating:** 4

**Summary:**

In this paper, the authors propose to use diffusion models to perform unsupervised anomaly detection on an in-house chest CT-scan dataset, with 2 types of non-healthy patients : COVID and ILD. They train a standard diffusion model to denoise steps of simplex noise (as in Wolleb et al 2022), but add a conditioning by a lung mask, obtained itself with another segmentation network. At inference they partially noise the image and denoise it to recover a pseudo healthy image (as in Wolleb et al 2022). At inference, they also adapt the threshold of each anomaly map based on the anomaly scores of the background, with the aim that the noise level of the anomaly scores in the background could inform on the rest of the anomaly map scores (e.g. higher std in background could mean that model is kind of lost). They also perform ensembling by pooling 7 different anomaly score maps obtained by noising with 7 different seeds. The authors prove that each improvement (3 + simplex noise) they propose increase performance on the 2 types of patients, and that they outperform some SOTA methods (diffusion, foundation-model).

**Strengths:**

- The paper is well written
- The experiments are clear and the results easy to understand
- The improvements proposed are simple
- The authors propose nice visualizations that are well commented
- The authors take the time to discuss their results and limits

**Weaknesses:**

- The bibliography is very focused on diffusion methods, it could be more general, or maybe the authors could precise that their bibliographic review has a strong focus on diffusion methods. See Ruff et al. review for a wide range of anomaly detection methods for instance.
- The novelty of the proposed improvement is quite modest. Simplex noise already exist for diffusion (Wolleb et al.), ensembling is a very common technique. Conditioning on a segmentation indeed seemed new. The background adaptative method is smart but quite simple and still relies on an hyperparameter to tune, moreover it imposes a binarization (a clinician could want an anomaly score map that he/she could threshold to its sensitivity/specificity liking).
- The method's heavily relies on another network's segmentation quality

**Detailed Comments:**

- "particularly within the domain of medical imaging (Wolleb et al., 2022)." I don't see the link between the citation and the claim
- "Therefore, rather than synthesizing abnormal images which often result in unrealistic or oversimplified pathologies, some works (Wolleb et al., 2022; Wyatt et al., 2022; Cai et al., 2025; Bercea et al., 2025) have turned to using generative models to learn the distribution" this first part talks about models that learn normal data and then latter on the authors discuss specifically diffusion models. However, in this first part 2 out of 4 methods cited are already diffusion methods. The reviewer believes that there are numerous methods that learns the normal data, even generative methods (Normalizing flows for instance, or VAE), and believe the authors could expand their scope to non-diffusion methods here (or even better non-generative !).
- The 2.2 paragraph could be entirely removed and replaced with one sentence, as the simplex noise is very popular for diffusion models in anomaly detection
- "Since the conditioning mechanism is agnostic to disease labels, it generalizes naturally to any unseen pathologies" : I would not say that because your conditioning mechanism strongly relies on the quality of the segmentation, which we do not know the capabilities of generalization
- "During inference (see Alg. 1), we apply a partial forward diffusion (Tsampling < Ttraining) to each query sample from the abnormal datasets. This ensures that the corrupted image preserves anatomical informatio" : this is what is done in every UAD diffusion paper, as it would make no sense to noise the image entirely. Can the authors rephrase this to not make it sound like a novelty ?
- "We further apply a fully connected conditional random field (Kr¨ahenb¨uhl and Koltun, 2012) to enforce spatial consistency and obtain the final anomaly mask." : this point is only stated once and never discussed again. Could the authors explain in more details this procedure ? How does it improve the quality of the anomaly score map ? Why is this improvement not in the result table? (why is it not a variant)
- Could the authors make sure that ensembling in diffusion models as not been done before ? Please cite the appropriate work if relevant, or clearly state that this is a novelty.
- "we used a fixed random seed for simplex noise generation across slices within the same 3D scan at inference time" : does this mean that the noise was the same for all slices ? If not then what does it do to fix the seed ? Can the authors elaborate ?
- Could the authors explain to the non-clinicians readers how in figure 4, in the axial view of the ILD, the grey mass that seem to obstruct the lung, is not an anomaly ?
- Can the authors make it more clear in the table and in the text that the variants are variants of SACdiff ?
- "These are the limitations of the current supervised methods and foundation model" : could you discuss the failure cases of these models ? Why is that the case ?
- "uncertainty-aware ensembling" is used to call an averaging of 7 maps, I don't see how a mean is an "uncertainty-aware ensembling" as the std is not used in this ensembling and the authors state that the std is a proxy for uncertainty.
- "offering insights into the model’s reliability and facilitating a more robust clinical interpretatio" : I think the authors could read about uncertainty and why std on different seed iteration is not quantifying entirely the uncertainty. The reviewer apologizes not to give any reference on this matter as he/she thinks he/she's not knowledgeable enough on that subject.
- " The arrows highlight regions where inherent consistency in ensembling leads to improved detectio" : as the std is not used, I don't see why the authors highlight region with high std stating that these regions will be improved by averaging.
- The whole paragraph on "Model Interpretability" in discussion is not a "future direction" as stated

**Justification Of Final Rating:**

The scope of the bibliography has been extended, and the authors spent a great deal of work answering the reviewer's concerns, especially with the segmentation network and overstated claims. I will update my rating to weak accept, given the modest novelty of the contributions.
The reviewer wants to thank the authors for their work and restate the strengths of this paper :
- The paper is well written
- The experiments are clear and the results easy to understand
- The improvements proposed are simple
- The authors propose nice visualizations that are well commented
- The authors take the time to discuss their results and limits

**Justification Of The Preliminary Rating:**

Given the modest novelty of the contributions, the narrow scope of the bibliography, and some important questions and comments the reviewer has, he/she believes he/she still need to be convinced as to accept or not this paper.

**Questions To Address In The Rebuttal:**

Please see the detailed comments above, the reviewer believes he/she did not put any comment that where particularly minor and should not be addressed

---

> ### Author Response · Authors · 2026-01-25
> **Response to Reviewer hpuJ**
>
> We would like to express our gratitude to the reviewer for their thoughtful evaluation of our manuscript. In response to the reviewer’s comments, we are pleased to have the opportunity to address the raised questions. These updates have been incorporated into the revised manuscript.
>
> **(1) "particularly within the domain of medical imaging (Wolleb et al., 2022)." I don't see the link between the citation and the claim**
>
> Thank you for raising this point. As noted in the referenced work, “In medical image analysis, pixel-wise annotated ground truth is hard to obtain, often unavailable, and contains a bias from human annotators,” we would like to emphasize that the scarcity of annotated data is a fundamental challenge in medical AD. This challenge therefore motivates the cited work and our method. We have added additional references to further support this claim and highlight that it is a pronounced problem in medical field: “particularly within the domain of medical imaging (Wolleb et al., 2022; Huang et al., 2023; Tschuchnig and Gadermayr, 2022).”
>
>
> **(2) "Therefore, rather than synthesizing abnormal images which often result in unrealistic or oversimplified pathologies, some works (Wolleb et al., 2022; Wyatt et al., 2022; Cai et al., 2025; Bercea et al., 2025)…" this first part talks about models that learn normal data and then latter on the authors discuss specifically diffusion models. However, in this first part 2 out of 4 methods cited are already diffusion methods. The reviewer believes that there are numerous methods that learns the normal data, even generative methods (Normalizing flows for instance, or VAE), and believe the authors could expand their scope to non-diffusion methods here (or even better non-generative !).**
>
> Thank you for raising this question. While our review focused on recent generative methods and emphasizes diffusion-based methods due to their direct relevance to our task and approach, we agree that broader literature is important and have now referred to the review by Ruff et al.
>
>
> In the initial submission, we discussed VAE- and GAN-based methods in the latter part of this paragraph (we have briefly mentioned several non-generative approaches in the second paragraph of the Introduction, but would like to emphasize unsupervised generative methods for medical images as these are most relevant to the goal addressed in this work). In the revised manuscript, we now cite these methods in the opening sentence as well, to improve clarity and flow. We have further **expanded the literature review to include additional non-diffusion approaches in order to provide a more comprehensive context**, including VAE, ceVAE, GANomaly, SteGANomaly, MADGAN, f-AnoGAN and some other models.
>
> The revised Introduction now reads: “Therefore, rather than synthesizing abnormal images which often result in unrealistic or oversimplified pathologies, some works (Zimmerer et al., 2019; Chen et al., 2019; Uzunova et al., 2019; Lu and Xu, 2018; Akcay et al., 2019; Han et al., 2021; Baur et al., 2020; Schlegl et al., 2019; Wolleb et al., 2022; Wyatt et al., 2022; Cai et al., 2025; Bercea et al., 2025) have turned to using generative models to learn the distribution of normal data. In this paradigm, models are trained to capture healthy anatomy, enabling the detection of any out-of-distribution anomalies at inference time as deviations from the learned manifold. This idea has been applied for VAE (Zimmerer et al., 2019; Chen et al., 2019; Uzunova et al., 2019; Lu and Xu, 2018) and GAN-based methods (Akcay et al., 2019; Schlegl et al., 2019; Baur et al., 2020; Han et al., 2021) showing initial success on natural and medical images.”
>
>
> **(3) The 2.2 paragraph could be entirely removed and replaced with one sentence, as the simplex noise is very popular for diffusion models in anomaly detection**
>
> Thank you very much for raising this concern. We appreciate the feedback. Since the noise discussed in **Section 2.2** is different from standard practice for diffusion models and not a mainstream technique on its own, we felt it was important to provide the motivation for this choice and context for readers from the broader field to ensure clarity and completeness. We expanded on power-law assumption and multi-octave design in this section in response to another reviewer, and in response to this comment, we have reduced the details of the noise and moved them to **Appendix C**. We hope this addresses your concern while preserving the integrity and clarity of the paper.

---

> ### Author Response · Authors · 2026-01-25
> **Response to Reviewer hpuJ**
>
> **(4)"Since the conditioning mechanism is agnostic to disease labels, it generalizes naturally to any unseen pathologies" : I would not say that because your conditioning mechanism strongly relies on the quality of the segmentation, which we do not know the capabilities of generalization**
>
> Thank you very much for raising this concern. We agree that the performance of lung-mask conditioning depends on the quality of the segmentation, and we would like to clarify that its generalization is disease-agnostic because it segments on anatomical structures rather than pathological features. Our intention was to emphasize that the segmentation network used in this work (Chaganti et al., 2020) has been validated across multiple datasets and disease types, and that the conditioning strategy itself is not restricted to any particular disease. We have revised the manuscript to better motivate and clarify this point, and now state: “Since the conditioning mechanism and segmentation network is agnostic to disease labels, it generalizes naturally to **any unseen disease types**.”
>
>
> **(5) "During inference (see Alg. 1), we apply a partial forward diffusion (Tsampling < Ttraining) to each query sample from the abnormal datasets. This ensures that the corrupted image preserves anatomical informatio" : this is what is done in every UAD diffusion paper, as it would make no sense to noise the image entirely. Can the authors rephrase this to not make it sound like a novelty ?**
>
> We thank the reviewer for this comment. Our intention was not to claim novelty, but to describe our implementation protocol. We have revised the sentence to read: “During inference (see Alg. 2), we apply a partial forward diffusion (Tsampling < Ttraining) to each query sample from the abnormal datasets. **As in current unsupervised diffusion-based AD works (Wang et al., 2024)}**, this ensures that the corrupted image preserves anatomical information…”
>
> **(6) "We further apply a fully connected conditional random field (Kr¨ahenb¨uhl and Koltun, 2012) to enforce spatial consistency and obtain the final anomaly mask." : this point is only stated once and never discussed again. Could the authors explain in more details this procedure ? How does it improve the quality of the anomaly score map ? Why is this improvement not in the result table? (why is it not a variant)**
>
> We thank the reviewer for raising this question. We observed that without CRF, the resulting anomaly maps were spatially discontinuous. The CRF is therefore a fundamental component of our pipeline for enforcing spatial consistency, readability and smoothness of the masks. When omitted, all baseline methods exhibited degraded performance; consequently, we applied the CRF uniformly to all baselines **as part of the pipeline** to ensure a fair and consistent comparison.
>
>
> **(7) Could the authors make sure that ensembling in diffusion models as not been done before ? Please cite the appropriate work if relevant, or clearly state that this is a novelty.**
>
> We thank the reviewer for the comment. The idea of using multiple draws and aggregation has been validated in prior works. Our approach adopts this concept in the context of diffusion models for anomaly detection and uses it to quantify uncertainty in anomaly masks. We have updated **section 2.5**: “The effectiveness of using multiple draws followed by aggregation has been validated in prior generative model–based studies (Whang et al., 2022; Ekmekci and Cetin, 2023). We adopts this concept, uses a multi-sample inference strategy within diffusion models for anomaly detection and leverage it to quantify uncertainty in the resulting anomaly masks.”
>
>
> **(8) "we used a fixed random seed for simplex noise generation across slices within the same 3D scan at inference time" : does this mean that the noise was the same for all slices ? If not then what does it do to fix the seed ? Can the authors elaborate ?**
>
> We thank the reviewer for the question. For each scan, a fixed random seed is used to generate the initial simplex noise, ensuring a consistent noise pattern across slices within the same 3D volume. This design stabilizes the reconstruction while still allowing variability across different scans. We have clarified this implementation detail in the Methods **section 3.2**: “To ensure slice consistency within each subject, a fixed random seed is used at inference time to generate the same initial simplex noise for each 3D volume, ensuring a consistent noise pattern across slices that stabilizes reconstruction.”

---

> ### Author Response · Authors · 2026-01-25
> **Response to Reviewer hpuJ**
>
> **(9) Could the authors explain to the non-clinicians readers how in figure 4, in the axial view of the ILD, the grey mass that seem to obstruct the lung, is not an anomaly ?**
>
> We thank the reviewer for the comment. The grey mass next to the lung corresponds to normal anatomical structures and does not represent a pathological pattern relevant to the anomalies of interest. It lies outside the lung fields, whereas ILD abnormalities occur within the lung and typically present as ground-glass opacities, reticulation, or honeycombing. The appearance of this grey mass is not consistent with ILD-related pathology.
>
>
> **(10) Can the authors make it more clear in the table and in the text that the variants are variants of SACdiff ?**
>
> We thank the reviewer for the comment. We labeled SAC-Diff variants in the table in the revised manuscript, so it is explicit which rows correspond to which component ablations in the current version.
>
>
> **(11) "These are the limitations of the current supervised methods and foundation model" : could you discuss the failure cases of these models ? Why is that the case ?**
>
> We thank the reviewer for raising this important point. The observed failure of supervised methods and foundation models primarily stem from their reliance on labeled training data and their sensitivity to domain shift. Supervised models are optimized for a narrow set of annotated pathologies and therefore tend to overfit to dataset-specific visual patterns and fail to generalize to unseen conditions. Foundation models, while trained on large datasets, are limited by the implicit assumptions learned during training. When applied to images acquired with different scanners, protocols, or exhibiting atypical abnormalities, these models may prone to confident but incorrect predictions under domain shift.
>
> We have updated this paragraph in **section 3.4** to include: “The failures of current supervised and foundation models largely stem from the reliance on labeled data and sensitivity to domain changes. These observations highlight these limitations, including limited generalization to unseen or heterogeneous pathologies and sensitivity to domain shifts arising from differences in scanners and acquisition protocols.”
>
> **(12) "uncertainty-aware ensembling" is used to call an averaging of 7 maps, I don't see how a mean is an "uncertainty-aware ensembling" as the std is not used in this ensembling and the authors state that the std is a proxy for uncertainty.**
>
> We thank the reviewer for the comment. To clarify, the voxel-wise standard deviation is used as a proxy for epistemic uncertainty, while ensembling in this context refers to averaging multiple maps to improve consistency and robustness. The standard deviation is not used as a weight during ensembling, it is computed after averaging to quantify uncertainty. Our approach leverages multiple samplings to produce a robust average for stable performance, with the post-ensembling standard deviation providing an interpretable measure of uncertainty. We hope this clarifies the point and appreciate the reviewer’s attention.

---

> ### Author Response · Authors · 2026-01-25
> **Response to Reviewer hpuJ**
>
> **(13) "offering insights into the model’s reliability and facilitating a more robust clinical interpretatio" : I think the authors could read about uncertainty and why std on different seed iteration is not quantifying entirely the uncertainty. The reviewer apologizes not to give any reference on this matter as he/she thinks he/she's not knowledgeable enough on that subject.**
>
> We thank the reviewer for the valuable comment. As noted in uncertainty quantification related work (He et al., 2025), although the variance (or standard deviation) does not capture all aspects of systematic uncertainty, it can be used as a proxy for prediction uncertainty. Model and data uncertainty can arise from multiple sources (e.g. uncertainty in data variability, in the choice of model family, in parameter learning etc.) and there are various ways to represent model uncertainty from each type. In our case, **the predictions of the ensemble form a distribution, therefore the standard deviation across these predictions provides an estimate of prediction uncertainty**.
>
> We have revised the text to state: “…standard deviation provides an estimate of epistemic **prediction uncertainty** (He et al., 2025)” in **section 2.5** and “the ensembling strategy enables uncertainty quantification for the model's predictions” in **section 3.4**, clarifying that the measure reflects **uncertainty in the model’s predictions**. We appreciate the reviewer’s insight and hope this clarification provides greater clarity.
>
>
> **(14) " The arrows highlight regions where inherent consistency in ensembling leads to improved detectio" : as the std is not used, I don't see why the authors highlight region with high std stating that these regions will be improved by averaging.**
>
> We thank the reviewer for the comment. Warmer colors indicate **low standard deviation** and **high model confidence**, while the arrows highlight regions where the ensembled maps show **improved results compared to individual sample predictions**. For example, in the first sample shown, a single-sample anomaly detection missed an anomaly, whereas the ensembled result clearly detected it with high confidence (pointed out by the purple arrow). These results demonstrate that ensembling reduces random variation and improves both robustness and reliability.
>
>
> **(15) The whole paragraph on "Model Interpretability" in discussion is not a "future direction" as stated**
>
> We thank the reviewer for the comment. We have revised “Our potential future directions are discussed below” to “We summarize key discussion points and outline potential directions for future work below” to improve the flow and structure of the paper.

---

> > ### Comment · Reviewer_hpuJ · 2026-01-28
> >
> > I would like to thank the authors for the discussion and clarifications. I will respond to each comment bellow. No response means I'm satisfied with the answer.
> >
> > 4)  "Our intention was to emphasize that the segmentation network used in this work (Chaganti et al., 2020) has been validated across multiple datasets and disease types" : does this claim comes from the original paper ? I think it should be stated very clearly and discussed because it is an important matter, i.e. please state in the text (if it is the case) that the segmentation network has been tested on such and such disease. Moreover, I haven't read the paper but if this network has been, say, tested on 5 different disease types, than the claim "it generalizes naturally to any unseen disease types" is overstated.
> >
> > 6) I think this finding is very interesting and should be presented in the main body. I understand that if the performances are too degraded it could not be presented in the table but should definitively be stated in the text. Now that you mentioned that the CRF is a crucial part of the pipeline I re-state my claim that this part of the procedure should be explained more extensively in the text.
> >
> > 8) I am sorry but I still did not understand what the authors meant. Is the noise pattern the same for each 3D subject ? Is it the same for each multiple draw of diffusion (see 7)) ? What do the authors mean by "slice consistency" ? What do the authors mean by "stabilizing reconstruction" ? What is a "consistent" noise pattern ? To make it cristal clear I think the authors should not use the word "consistent" or "consistency" as it is subject to very diverse interpretations
> >
> > 9) I'll let the authors decide but I would be inclined to think that this information should be present in the legend or text.
> >
> > 12) I completely agree with the authors, which leads me to confirm that calling an average of 7 maps  "uncertainty-aware ensembling" is an overstatement. I would advise the authors to not oversell this point.
> >
> > The reviewer thinks that the authors have addressed a large part of the reviewer's concerns in a pretty serious way and wants to thank the authors for the quality of their response. I'll be likely updating my rating to weak accept by the end of the discussion.

---

> > > ### Author Response · Authors · 2026-01-29
> > > **Authors' Response for Reviewer hpuJ**
> > >
> > > **(8) I am sorry but I still did not understand what the authors meant. Is the noise pattern the same for each 3D subject ? Is it the same for each multiple draw of diffusion (see 7)) ? What do the authors mean by "slice consistency" ? What do the authors mean by "stabilizing reconstruction" ? What is a "consistent" noise pattern ? To make it cristal clear I think the authors should not use the word "consistent" or "consistency" as it is subject to very diverse interpretations**
> > >
> > > Thank you for the question. To clarify, we use a fixed random seed to generate one initial noise for all 2D slices in a 3D volume during inference. This ensures that **initial noise is the same across slices for a single 3D volume inference**, but it will be different for another independent inference. Slice consistency means maintaining structural stability across generated 2D slices, accordingly, by stabilizing reconstruction, we refer to the reconstructed image having structural stability. While we understand that the term consistency can be interpreted in multiple ways, this concept has been referred to as **slice consistency** and framed as the **slice inconsistency problem** (also called **inter-slice** or **slice-by-slice inconsistency problem**) in prior works (Chen et al., 2025; Li et al., 2024; Fang et al., 2024), which is why we use the term here.
> > >
> > >
> > > Tianqi Chen et al. 2.5D Multi-View Averaging Diffusion Model for 3D Medical Image Translation: Application to Low-Count PET Reconstruction With CT-Less Attenuation Correction. IEEE Transactions on Medical Imaging, 44(11):4239–4250, November 2025. ISSN 1558-254X. doi: 10.1109/TMI.2025.3570342.URL https://ieeexplore.ieee.org/abstract/document/11005585.
> > >
> > > Zirong Li et al. Two-and-a-half order score-based model for solving 3D ill-posed inverse problems. Computers in Biology and Medicine, 168:107819, January 2024. ISSN 1879-0534. doi: 10.1016/j.compbiomed. 2023.107819.
> > >
> > >
> > > Wei Fang et al. CycleINR: Cycle Implicit Neural Representation for Arbitrary-Scale Volumetric Super-Resolution of Medical Data. In 2024 IEEE/CVF Conference on Computer Vision and Pattern Recognition (CVPR), pages 11631–11641, Seattle, WA, USA, June 2024. IEEE. ISBN 9798350353006.
> > >
> > > **(9) I'll let the authors decide but I would be inclined to think that this information should be present in the legend or text.**
> > >
> > > Thank you very much for the suggestion. We considered including it in the figure legend but found that including this information in the figure legend or main text may be somewhat redundant as we already presented the annotation map to show where anomaly and normal regions are. We sincerely appreciate the reviewer’s thoughtful guidance and have given this comment careful consideration.
> > >
> > > **(12) I completely agree with the authors, which leads me to confirm that calling an average of 7 maps "uncertainty-aware ensembling" is an overstatement. I would advise the authors to not oversell this point.**
> > >
> > > We thank the reviewer for pointing this out. We acknowledge that the term “uncertainty-aware ensembling” could be misleading. In the revision, we renamed this component to **“multi-sample ensembling with uncertainty estimation”** and clarify that uncertainty is **not directly guiding the ensembling** itself, but is estimated **post hoc** via the voxel-wise standard deviation across stochastic reconstructions. In Section 3.4, “In addition, **multi-sample ensembling with uncertainty estimation** further improves the model's anomaly detection performance...” In Figure 6, “Examples of automatic anomaly detection results with **ensembling** enabled.”

---

> > > > ### Comment · Reviewer_hpuJ · 2026-01-29
> > > >
> > > > I would like once again to thank the authors for the discussion and clarifications. I will respond to each comment bellow. No response means I'm satisfied with the answer.
> > > >
> > > > 6) I want to thank the authors for having provided the figure in appendix E and clarifications in the main body. I still think that the CRF, as it is non-trivial and essential in the pipeline (this point is even more valid after seeing the new figure) should be explained more and motivated : why did you use it ? how does it work ? And not only state that it is applied to all methods.
> > > >
> > > > 8) The reviewer wants to thank the authors for the clarification. I strongly think that the text should be updated in the main body, by taking elements for your response, to be unambiguous and cristal clear to every reader. I let the authors decide how they should or should not update the main body text.
> > > >
> > > > 12) The reviewer want to thank the authors for the modifications. Sorry for being insistant about this again but the authors state that "in addition, multi-sample ensembling with uncertainty estimation further improves the model’s anomaly detection performance", however, in the reported anomaly detection performances _no uncertainty estimation comes into play_. I believe the authors should let the "uncertainty part" exclusively in their qualitative analysis but not in their quantitative one !
> > > >
> > > > I will be updating my rating to weak accept but strongly encourage the authors to take into account these last comments, especially 6 and 12.
> > > > The reviewer wants to thank the authors for the quality of this discussion.

---

> > ### Author Response · Authors · 2026-01-29
> > **Authors' Response for Reviewer hpuJ**
> >
> > We thank the reviewer again for the thoughtful suggestions and detailed feedback. We are pleased to address the reviewer’s questions and comments, and the corresponding revisions have been incorporated into the manuscript to further strengthen the paper.
> >
> >  **(4) "Our intention was to emphasize that the segmentation network used in this work (Chaganti et al., 2020) has been validated across multiple datasets and disease types" : does this claim comes from the original paper ? I think it should be stated very clearly and discussed because it is an important matter, i.e. please state in the text (if it is the case) that the segmentation network has been tested on such and such disease. Moreover, I haven't read the paper but if this network has been, say, tested on 5 different disease types, than the claim "it generalizes naturally to any unseen disease types" is overstated.**
> >
> >
> > Thank you very much for the comment. We agree this is a very important point. The segmentation network was first trained on large-scale datasets (8087 patients, 8006 for training and 81 for validation), then fine-tuned on 1136 patients ( trained on 1076 and validated on 60) with multiple abnormality patterns including ground-glass opacities, consolidation, effusions, masses, and others to ensure robustness in abnormal scans. We have added these details of the lung segmentation framework to **Appendix A**: “For lung mask segmentation, the network was firstly trained on large-scale datasets (8087 patients, 8006 for training and 81 for validation), and then fine-tuned on 1136 patients (trained on 1076 and validated on 60) with multiple abnormality patterns, including ground-glass opacities, consolidation, effusions, masses, and others to ensure robustness in abnormal scans.” Regarding the claim, we revised it to be precise for the scope in **Section 2.3**: “Since the conditioning mechanism is agnostic to disease labels, and the segmentation network has been trained across multiple datasets and disease types in prior work (Chaganti et al., 2020), **it demonstrates reasonable robustness to anatomical and pathological variation**.”
> >
> >
> > **(6) I think this finding is very interesting and should be presented in the main body. I understand that if the performances are too degraded it could not be presented in the table but should definitively be stated in the text. Now that you mentioned that the CRF is a crucial part of the pipeline I re-state my claim that this part of the procedure should be explained more extensively in the text.**
> >
> > Thank you very much for the suggestion. We added a visualization to demonstrate the post-processing effect of CRF to **Appendix E**.  For completeness, we stated the finding in **Section 2.4**: “A fully connected conditional random field (Krahenbuhl and Koltun, 2012) is further applied to enforce spatial consistency, readability and smoothness of the anomaly masks (See Appendix E). **As part of the pipeline, this is uniformly applied to all baselines and our methods** to ensure a fair comparison.”

---

> ### Author Response · Authors · 2026-01-30
> **Authors' Response for Reviewer hpuJ**
>
> We sincerely thank the reviewer for the follow-up comments and the positive reassessment of our work.
>
>  - CRF post-processing is indeed a non-trivial and important component of our pipeline. As the deadline for re-uploading a revised manuscript has passed, we will further clarify its implementation and expand its motivation and mechanism in the camera-ready version.
>
>  - We also appreciate the reviewer’s suggestion to include the description of the gray mass in the main text. We will update the manuscript to ensure the presentation is clear for all readers.
>
>  - Regarding ensembling, we thank the reviewer for the careful observation. To avoid confusion, we will restrict references to uncertainty estimation in the camera-ready version.
>
> We thank the reviewer again for the constructive discussion, encouraging evaluation and the helpful suggestions, which we believe will further improve the clarity and rigor of the manuscript.

---

> > ### Comment · Reviewer_hpuJ · 2026-01-30
> >
> > Thanks for taking into consideration the suggestions and thanks again for the great work.

---

### Official Review · Reviewer_xk8z · 2026-01-11

**Confidence:** 4
**Preliminary Rating:** 3
**Final Rating:** 4

**Summary:**

This paper proposes SAC-Diff, an unsupervised anomaly detection framework for chest CT. The method trains a conditional diffusion model on control/healthy CTs, using multi-octave simplex noise for the forward diffusion perturbation and a lung mask as conditioning. At inference, a test CT is partially diffused for a chosen number of steps and then denoised to generate a pseudo-normal reconstruction. Anomalies are detected from a reconstruction error map (MSE between the input and the generated pseudo-normal image), followed by a scan-aware background-adaptive thresholding strategy (mean + λ·std computed from voxels outside the lung mask) and optional spatial regularization (e.g., CRF). The paper also emphasizes robustness/interpretability via multi-sample ensembling, producing a mean reconstruction and a voxel-wise variance/uncertainty map.

Overall, the reported results look decent and the “scan-aware” perspective is appealing, but several methodological choices require stronger justification and targeted ablations.

**Strengths:**

* Clear problem framing: Unsupervised anomaly detection for screening is clinically relevant and avoids reliance on dense annotations for every pathology.

* Reasonable technical contributions: simplex-noise diffusion (detail-preserving perturbation), conditioning with anatomical priors (lung masks), scan-specific background-aware thresholding, and consistency/uncertainty via multi-sample ensembling form a coherent pipeline.

* Evaluation on multiple disease cohorts (e.g., COVID-19 and ILD) with segmentation-style metrics (IoU/Dice) is a good start, and comparisons to several baselines/variants are helpful.

**Weaknesses:**

* Role and necessity of the reconstruction/MSE objective for detection is under-motivated, and it is unclear whether any additional L2 term is used in training beyond the standard diffusion objective. Since the framework’s anomaly score is explicitly 𝐸(𝑥)=∥𝑥−𝑥_0∥^2 (reconstruction error), the paper should justify why this is the most appropriate scoring signal versus diffusion-native alternatives (e.g., score-based likelihood proxies, denoising residuals at multiple timesteps, or energy aggregated over steps).

* Mask conditioning is implemented via early fusion (channel concatenation). This is simple but may not be optimal for leveraging spatial priors throughout the denoiser at multiple resolutions.

* Inference hyperparameters (notably the noising/sampling steps 𝑇_𝑠 and ensembling size K) appear empirically chosen. The sensitivity of performance/cost to these choices needs to be characterized.

* Distributional claims about “control-like” reconstructions would benefit from explicit embedding/feature-space analysis rather than only pixel-space error maps.

* Robustness to domain shift (scanner vendor, reconstruction kernel, slice thickness, site/protocol) remains insufficiently validated. Even with background-adaptive thresholding, the anomaly score and thresholding can still drift across centers.

**Detailed Comments:**

### (1) Why use an L2/MSE between input and generated image?
Please provide a clearer motivation: why is pixel-space reconstruction error during training given that the diffusion model is learning the small denoising steps that gradually projecting from the noise distribution to the data distribution. An ablation study should be conducted on this.

### (2) Is early fusion of the lung mask optimal?
The method concatenates the binary lung mask with the noised image as conditioning. Prior conditional generation/segmentation literature often injects masks via SPADE / FiLM-like modulation (multi-layer, multi-scale), which can be more expressive than input concatenation.

### (3) How is the optimal inference diffusion/noising step 𝑇_𝑠 determined?
The paper sets 𝑇_{training} and 𝑇_{sampling}​ based on “empirical results.” This is a key hyperparameter that trades off anomaly removal vs structure preservation. A a sensitivity curve of performance vs 𝑇_𝑠 (and ideally runtime) can be helpful for illustration.

### (4) Show that generated pseudo-normal samples match the control distribution (feature-space evidence).
I suggest adding a t-SNE/UMAP (or similar) plot using embeddings from (a) a frozen encoder (e.g., self-supervised CT feature extractor) or (b) intermediate UNet features, including: controls, abnormal inputs, and reconstructions. The goal is to demonstrate that reconstructions move abnormal samples toward the control manifold rather than creating a distinct “artifact manifold.” This would strengthen the paper’s core claim that reconstructions are “control-like.”

### (5) Robustness of thresholding under domain shift (protocol/center/scanner).
Even though the method uses background-adaptive thresholds (computed from voxels outside the lung mask), the reconstruction error distribution can still vary substantially across scanners and protocols. Evaluate cross-domain generalization (different center/protocol if available). If not, create a synthetic shift by perturbing controls with realistic CT variations (e.g., intensity scaling/offset, noise injection, blur/sharpening approximating kernels, slice-thickness simulation, mild ring artifacts), and quantify how stable the thresholding and detection performance remain.

**Justification Of Final Rating:**

The authors have provided sufficient clarification with their extended visualizations to show the effectiveness of the proposed method and basically addressed my concerns. I think the current manuscript is good enough for more insight discussion in the conference.

**Justification Of The Preliminary Rating:**

I lean Borderline because the paper presents a coherent diffusion-based pipeline with incremental but plausible improvements, yet several key design choices and validation gaps prevent me from confidently recommending acceptance.

**Questions To Address In The Rebuttal:**

Please refer to the “Weaknesses / Major concerns” and “Detailed comments and questions (1)–(5)” sections above.

---

> ### Author Response · Authors · 2026-01-25
> **Authors' Response for Reviewer xk8z**
>
> We would like to thank the reviewer for providing valuable feedback. We are pleased to address the questions and conduct additional analyses to respond to the reviewer’s thoughtful comments. These updates have been incorporated into the revised manuscript to further strengthen the paper.
>
> **(1) Why use an L2/MSE between input and generated image?**
>
> Thank you for pointing out this question. We would like to clarify a key point that may have caused the confusion. During training, our model follows the standard formulation and is optimized using an L2 loss on the predicted noise, rather than the reconstruction between input and generated image. The training objective is not pixel-space reconstruction but the standard simplified ELBO loss. We have updated **figure 1(a)** and **section 3.2** to make this explicit: “The training objective was the L2 noise-prediction loss, which corresponds to optimizing a simplified variational evidence lower bound.”
>
> **(2) Is early fusion of the lung mask optimal? SPADE/FiLM-like modulation (multi-layer, multi-scale) can be more expressive.**
>
> Thank you very much for raising this insightful question. We agree that more expressive conditioning mechanisms can be powerful and effective, particularly in semantic image generation tasks where the conditioning signal consists of dense semantic labels encoding rich multi-class information. In our setting, the lung mask serves as **a binary coarse anatomical prior** and our goal is to use the lung mask to **restrict the model to anatomically plausible regions, rather than to prescribe detailed semantic content**. Given the low-dimensional and geometric nature of this conditioning signal, early fusion via channel concatenation is **intuitive and empirically stable** for our task. To compare conditioning mechanisms, we have added qualitative results in **Appendix D** from earlier experiments comparing reconstruction outcomes using cross-attention, SPADE, and channel concatenation. We have further clarified this design choice and explicitly note other conditioning methods as a promising direction for future work **in discussion**:
>
> “While early fusion is empirically efficient and stable in our case, more expressive conditioning mechanisms, such as cross-attention, SPADE or FiLM-based feature modulation, can enable richer interactions between $x_t$ and $c$. As shown in Appendix D, given the low-dimensional and geometric nature of our conditioning signal, early fusion via channel concatenation is empirically sufficient for our task when compared with SPADE and cross-attention. We leave a more comprehensive investigation of alternative conditioning strategies for future work.
>
> **(3) How is the optimal 𝑇_𝑠 determined?**
>
> We appreciate the reviewer’s insightful comment. Tₛ is indeed a key hyperparameter controlling the trade-off between anomaly suppression and structural preservation. Following prior literatures (Wyatt et al., 2022, Wang et al., 2024), the partial diffusion step in inference is chosen when **disrupted of anomaly distribution is observed but anatomical structure is still preserved**. In our observations, performance is good over a range of Tₛ values so we chose Tₛ = 550 for both performance and time consideration. We would like to present sensitivity curves showing how IoU and DICE scores vary with Tₛ for both testing datasets in **Appendix E**.
>
> **(4) Show that generated pseudo-normal samples match the control distribution (feature-space evidence).**
>
> We thank the reviewer for this valuable suggestion. We focused on pixel-space and task-relevant evidence in the first submission as our goal was anomaly localization and delineation in image space, and the pseudo-normal reconstructions serve as an intermediate production. However, we fully agree that feature-space analysis provides complementary insight to support our interpretation. We therefore extracted features from healthy controls, abnormal inputs, and reconstructed images from abnormal scans and generated UMAP embeddings to visualize these three groups. While we did not observe disease-class separated clusters in the embedding space, the plot indicates that **reconstructed images do not form a distinct manifold**. To further quantify this, we computed the feature-space distances between groups: **Healthy→Abnormal: 1.1371 and Healthy→Reconstructed pseudo-normal: 1.1135**. We have added these results to **Appendix F** and noted it in **section 3.3**:
>
> “In Appendix F we show a UMAP of features extracted from healthy, unhealthy, and model-reconstructed pseudo-healthy samples using a foundation model. Feature-space distances were further computed, yielding 1.1371 between healthy and unhealthy samples and 1.1135 between healthy and reconstructed pseudo-healthy samples.”

---

> > ### Comment · Reviewer_xk8z · 2026-02-01
> >
> > Thanks for the detailed response. The extended experiments have sufficiently illustrated that the effectiveness of the proposed method.

---

> ### Author Response · Authors · 2026-01-25
> **Authors' Response for Reviewer xk8z**
>
> **(5) Robustness of thresholding under domain shift (protocol/center/scanner).**
>
> Thank you very much for the insightful question! We agree that cross-scanner, cross-center, and cross-protocol robustness is critical for medical AD tasks. While multi-center data are not available to us, we would like to clarify that **our dataset is already cross-domain**, includes images acquired from **different scanners, reconstructed using different parameters, imaging protocols**, and containing both **contrast and non-contrast variations**. We added detailed descriptions of the dataset in **Appendix G**, listing the different scanners and protocols used in this study. We also emphasized this point in the revised manuscript in **section 3.1**:
>
> “For the generalization and robustness of the model, both training and testing datasets are heterogeneous, including images acquired from different scanner manufacturers, imaging protocols, reconstructed using varying parameters, containing both contrast-enhanced and non-contrast scans. A list of detailed descriptions of the dataset in Appendix G.”

---

### Author Rebuttal · Authors · 2026-01-25

**Rebuttal:**

We thank the reviewers for their constructive feedback. We are encouraged that reviewers highlighted the strengths of our work SAC-Diff, including the clear problem formulation, reasonable technical contributions of detail-preserving simplex noise perturbation, anatomical prior conditioning, scan-aware thresholding, and consistency-enhanced ensembling, as well as strong validation on COVID and ILD cohorts with extensive ablations.

In response to all suggestions from the reviewers, we revised the manuscript to better justify key design choices, strengthen empirical validation, and clarify the methodological contributions of SAC-Diff. The major concerns are addressed as below:

• Novelty clarification. SAC-Diff is not a collection of incremental tricks, but **a principled scan-aware formulation** of diffusion-based anomaly detection that **addresses anatomy, heterogeneity, and uncertainty in a unified framework**. Anatomical conditioning enforces structural priors that are essential in medical imaging but absent from general diffusion AD; background-adaptive thresholding addresses distributional shift and heterogeneous anomalies; ensembling enhances performance, consistency and interpretability. These were not jointly addressed in previous works.

• Anatomical conditioning. We added comparisons between early fusion, SPADE, and cross-attention. Given the low-dimensional geometric nature of the lung-mask prior, early fusion is empirically sufficient and stable; richer conditioning is left for future work.

• Feature-space validation. We added UMAP visualizations and feature-distance analysis using foundation-model embeddings, showing that reconstructed abnormal scans move toward the healthy manifold rather than forming an artificial cluster.

• Hyperparameter selection. We discussed stability across a range of T_sampling, and motivated the chosen value based on the anomaly-suppression vs. structure-preservation trade-off.

• Ensembling with uncertainty estimation. We clarified that voxel-wise variance is a proxy for epistemic uncertainty used for interpretation, while averaging multiple reconstructions improves robustness and consistency.

• Literature review. We expanded the review to include AE-, GAN-, and flow-based unsupervised AD methods and sharpened SAC-Diff’s positioning as a unified, scan-aware diffusion framework.

Please see the revised manuscript and point-by-point responses. We believe this paper is now well aligned with MIDL’s emphasis.

**Supporting Material:**

/attachment/02a8bde60a59cbd816029a29fdd57b8e86d85e6b.pdf

---

### Author Response · Authors · 2026-02-02
**Further Clarification on Reproducibility and Formalization**

We sincerely thank all the reviewers’ constructive feedback. In response to Reviewer iTgv’s final reviews, we would like to clarify two remaining points regarding reproducibility and methodological formalization.

**(1) Reproducibility.** We fully acknowledge the importance of reproducibility. As this work was conducted in an **industrial institution**, we are not permitted to share data or code directly with the public. However, we did take several steps to maximize reproducibility within these constraints:

• We provide **detailed implementation descriptions**. All pre-processing steps, model components, model hyperparameters, diffusion schedules, training and inference procedures, and ablation settings are explicitly reported in the manuscript or appendix;

• While original data cannot be shared directly, we are open to providing **access through research collaboration agreements**, subject to institutional and regulatory approvals;

• We are developing **a prototype for future release** to run the SAC-Diff pipeline on non-proprietary data.

We believe this enables meaningful reproducibility of the proposed method within appropriate privacy and legal frameworks.


**(2) Improved Formalization.**
In the revision, we have further strengthened the technical clarity by:

• We explicitly defined all diffusion variables and notations in the updated manuscript and appendix (including $T_{training}$ and $T_{sampling}$);

• We formally specified the training and sampling objective and algorithms, as well as multi-octave simplex noise formalization and anatomical conditioning mechanism;

• We clarified the power-law assumption underlying the noise design with supporting literature to prior work.

These additions make the algorithmic formulation and inference pipeline **fully explicit and implementable** from the paper alone. We believe the manuscript demonstrates its substantial methodological rigor to contribute to the MIDL community.

---

### Meta-Review · Area_Chair_BEuW · 2026-02-09

**Recommendation:** Accept (Poster)
**Confidence:** 4

**Metareview:**

This manuscript introduces a denoising diffusion probabilistic model (DDPM)–based method designed to improve the precision and sensitivity of anomaly detection by accounting for patient-specific anatomical and structural variability. The work’s strengths include the clear clinical relevance of enhanced anomaly detection for radiologists, the practical implementation of a DDPM-based framework supported by strong experimental results, and a logically structured presentation. Although the authors have provided thorough responses to the reviewers’ questions, many of which led to clarifications in the revised manuscript, reviewers remain concerned that the contribution is largely incremental and offers limited methodological novelty.

---

### Decision · Program_Chairs · 2026-02-13

Accept (Poster)